# Interpretable Kernel Representation Learning at Scale: A Unified Framework Utilizing Nyström Approximation

## Abstract

Kernel methods provide a theoretically grounded framework for non-linear and non-parametric learning, with strong analytic foundations and statistical guarantees. Yet, their scalability has long been limited by prohibitive time and memory costs. While progress has been made in scaling kernel regression, no framework exists for scalable kernel-based representation learning, restricting their use in the era of foundation models where representations are learned from massive unlabeled data. We introduce KREPES—a unified, scalable framework for kernel-based representation learning via Nyström approximation. KREPES accommodates a wide range of unsupervised and self-supervised losses, and experiments on large image and tabular datasets demonstrate its efficiency. Crucially, KREPES enables principled interpretability of the learned representations, an immediate benefit over deep models, which we substantiate through dedicated analysis.

## 1 Introduction and Related Works

Representation learning aims to uncover meaningful low-dimensional representations that generalize across tasks by capturing underlying data structure in a compact, expressive form. Earlier approaches relied on handcrafted features, e.g., SIFT, SURF (Park et al., 2022) for images or bag-of-words for text, which were domain-specific and required expert knowledge. In contrast, learned representations optimized directly from data have become dominant for their adaptability. The advent of unsupervised and self-supervised learning (SSL) further advanced representation learning in settings with scarce or no labels, making SSL a central paradigm for pre-training foundation models (Devlin et al., 2019; Caron et al., 2021; Arik & Pfister, 2021), where pretext tasks or augmentations drive the discovery of meaningful features. Current practices in representation learning primarily focus on deep neural network based approaches. Yet, in the broader scope of data science, especially where interpretability or analytical foundations are key considerations, more traditional models continue to hold significant value across domains. In particular, decision trees, random forests, and kernel methods remain competitive in several learning problems (Shwartz-Ziv & Armon, 2022; Ghorbani et al., 2020), while also providing interpretable predictions (Sagi & Rokach, 2020; Aristoff et al., 2024). Specifically, kernel methods offer unique advantages: (i) they are non-parametric approaches that are expressive enough to arbitrarily approximate any nonlinear function (Micchelli et al., 2006); (ii) owing to the representer theorem, kernel models provide intuitive outputs by leveraging pairwise similarities between data points (Schölkopf et al., 2001); (iii) a rich theory of reproducing kernels allows one to derive strong statistical guarantees for various tasks such as regression (Mallinar et al., 2022), classification (Cui et al., 2023), clustering (Vankadara & Ghoshdastidar, 2020), representation learning (Blanchard et al., 2007), etc; (iv) beyond predictive modeling, the representer theorem can be leveraged for interpretability. Yeh et al. (2018) introduced Representer Point Selection, which leverages representer theorem to explain supervised predictions of neural networks. However, they employ representer theorem as a tool for post-hoc explanations of a trained network, and are limited to supervised tasks, leaving interpretable representations using representer theorem in self-supervised settings unexplored.

Representation learning via Kernel Principal Component Analysis (KPCA) (Sterge & Sriperumbudur, 2022) is known but is less used now. Shah et al. (2022); Esser et al. (2024) propose kernel-based SSL models but leave scalability unaddressed. Nevertheless, kernel theory remains central

to both theory and development of SSL and foundation models; e.g., kernel SSL has been used to link representations to spectral decomposition of kernel matrices, analyze the role of augmentation and architecture in generalization, and characterize the stepwise dynamics of SSL (Cabannes et al., 2023; Simon et al., 2023), raising motivation to explore whether kernels can be adapted for practical self-supervised representation learning.

Current SSL methods are driven by their ability to leverage vast amounts of unlabeled data to learn generalizable representations (Radford et al., 2021; Caron et al., 2021). However, this dependence on data volume poses a fundamental challenge for kernel models. Kernels are computationally expensive, specifically during training: for $n$ training samples, $O(n^2)$ memory is needed to store the kernel matrix, which is the kernel function $k(x, x')$ evaluated for every pair in dataset. Also, training can take up to $O(n^3)$ run-time, as it typically involves inversion or eigen decomposition of the kernel matrix. Subsequent predictions per test sample also requires $O(n)$ memory and runtime to compute the kernel similarities between test data and all training samples. While current self-supervised learning frameworks rely on massive amount of data, naïve kernel methods cannot scale to such regimes. This fundamental mismatch raises the following open question:

*Can we design scalable kernel-based self-supervised learning methods that bridge the gap between the theoretical advantages of kernels and the data-hungry demands of modern foundation models?*

Various approaches have been suggested to speed up kernel machines, primarily relied on approximation techniques such as Random Fourier Features (Rahimi & Recht, 2007) and Nyström methods (Williams & Seeger, 2000), with the latter often achieving better empirical accuracy (Yang et al., 2012). Scalable implementations like NYTRO (Camoriano et al., 2016), FALKON (Rudi et al., 2017), and EigenPro (Ma & Belkin, 2019; Abedsoltan et al., 2023) combine these approximations with efficient solvers to handle large datasets—but almost exclusively for kernel ridge regression. Although some theoretical works study the statistical performance of approximations for KPCA (Sterge & Sriperumbudur, 2022), we are unaware of any practical implementation focusing on self-supervised kernel-based representation learning at scale, leaving a gap our work addresses.

**Main Contributions.** Motivated by the lack of practical tools for large-scale kernel-based representation learning and the need for massive data in self-supervised foundation models, we introduce KREPES: a scalable framework supporting diverse kernels and a wide range of unsupervised and self-supervised loss functions. Our contributions are:

**(i)** Unified kernel formulation for representation learning. We provide the kernel-based formulation for a comprehensive list of loss functions, including simple and spectral contrastive, SimCLR (Chen et al., 2020), VICReg (Bardes et al., 2022), Barlow twins (Zbontar et al., 2021), BYOL (Grill et al., 2020), while incorporating Tikhonov regularization. KREPES further allows one to incorporate additional custom-defined loss functions.

**(ii)** Scalable Nyström-based optimization. We combine Nyström approximation with gradient-based optimization, allowing arbitrary loss minimization—unlike prior works restricted to kernel ridge regression. We suggest strategies for two key aspects of Nyström methods: landmark selection and preconditioning technics, which enable efficiency across diverse losses.

**(iii)** Empirical validation. We study the performance of KREPES on standard benchmarks, demonstrating that KREPES can handle large datasets. We find KREPES performs similar to ResNet and attention-based models while using much fewer parameters.

**(iv)** Interpretability via representer landmarks. As an immediate benefit of our kernel SSL framework, we provide interpretation of learned representations through a proposed influence score and analyze the conceptual profile of samples with concept activation vectors.

**Distinction to Neural Kernel Literature.** Existing kernel computation efforts, such as Neural Tangent Kernels (NTKs) (Jacot et al., 2018; Arora et al., 2019), demonstrate that kernels for infinitely wide networks can be computed on datasets like CIFAR-10, but they focus exclusively on supervised prediction and remain limited to relatively small datasets ($\leq$60,000 samples). Moreover, the computational cost is prohibitive—e.g., Novak et al. (2019) report ∼1,000 GPU hours (40 days on a single GPU) to compute the CIFAR-10 kernel. Compositional kernel frameworks, such as Convolutional Kernel Networks (Mairal et al., 2014) and their End-to-End variant (Mairal, 2016), as well as Neural Kernels Without Tangents (NKWT) (Shankar et al., 2020), attempt to build deep kernel analogues to convolutional networks. While these methods are valuable, they also train only with su-

pervised objectives, are restricted to modest-scale datasets, and in the case of NKWT, incur $\tilde{O}(n^2 d^2)$ complexity (n, d being size of the data and its dimension), leading to $\sim$1,000 GPU-hour computations even for small images. In contrast, our focus is on efficient large-scale kernel representation learning, meaning even previous neural kernels can be used as the kernel in our backbone.

## 2 REPRESENTATION LEARNING WITH KERNELS

We consider a framework with $n$ unlabeled samples, along with possible augmented views, from an input space $\mathcal{X}$. In the dataset $\{(x_i^1, \ldots, x_i^p)\}_{i=1,\ldots,n}$, each tuple $(x_i^1, \ldots, x_i^p)$ refers to $p$ augmentations associated with the $i$-th unlabeled sample. In unsupervised learning, $p = 1$ since only the unlabeled sample is observed. Self-supervised learning (SSL) requires $p \geq 2$.

The goal of representation learning is to learn an embedding $f : \mathcal{X} \to \mathbb{R}^h$ for some specified dimension $h$. Given the unlabeled (augmented) samples $\{(x_i^1, \ldots, x_i^p)\}_{i=1,\ldots,n} \subset \mathcal{X}^{np}$, this is achieved by minimizing a specified loss function $\mathcal{L}$:

$$\min_{f \in \mathcal{F}} \mathcal{L}\left(\left\{f(x_i^j)\right\}_{i \in [n], j \in [p]}\right), \tag{1}$$

where $\mathcal{F}$ denotes a class of functions, governed by the choice of machine learning models with neural network architectures being the typical choice (Bardes et al., 2022). In this work, we consider $\mathcal{F}$ to be the class of kernel models (Schölkopf et al., 2001). Precisely, let $k : \mathcal{X} \times \mathcal{X} \to \mathbb{R}$ be a real-valued positive definite kernel function with the associated Reproducing Kernel Hilbert Space (RKHS) $(\mathcal{H}, \langle \cdot, \cdot \rangle)$, and $\phi_x \in \mathcal{H}$ denotes the non-linear feature map corresponding to the input sample $x \in \mathcal{X}$. We recall that $k(x, x') = \langle \phi_x, \phi_{x'} \rangle$ for all $x, x' \in \mathcal{X}$. We assume that the loss $\mathcal{L}$ is optimized over

the function class $\mathcal{F} = \left\{ f(x) = \begin{bmatrix} \langle w_1, \phi_x \rangle \\ \vdots \\ \langle w_h, \phi_x \rangle \end{bmatrix}, w_1 \ldots w_h \in \mathcal{H} \right\}$. For convenience, we express any

$f(x) = W\phi_x$ with the understanding that the learnable map $W : \mathcal{H} \to \mathbb{R}^h$ corresponds to the inner products of $\phi_x$ with the learnable parameters $w_1, \ldots, w_h \in \mathcal{H}$, which are the rows of $W$.

In general, $\mathcal{H}$ can be infinite dimensional, making the optimization of $W$ computationally intractable. However, the representer theorems (Kimeldorf & Wahba, 1970; Schölkopf et al., 2001) allow one to rewrite equation 1 as a finite-dimensional optimization over $\mathbb{R}^{h \times np}$. It is useful to recall the insight behind the representer theorems for the subsequent discussion on efficient approximation for kernel methods. One can express any $f \in \mathcal{F}$ as $f(x) = W\phi_x = \overline{W}\phi_x + W^\perp\phi_x$ so that each row of $\overline{W}$ lies in the span of $\{\phi_{x_i^j}\}_{i \in [n], j \in [p]}$ and each row of $W^\perp$ is orthogonal to every $\phi_{x_i^j}$, that is, $W^\perp\phi_{x_i^j} = 0$ for all $i, j$. It immediately follows that $\mathcal{L}\left(\left\{f(x_i^j)\right\}_{ij}\right) := \mathcal{L}\left(\left\{W\phi_{x_i^j}\right\}_{ij}\right) = \mathcal{L}\left(\left\{\overline{W}\phi_{x_i^j}\right\}_{ij}\right)$, which implies that it is sufficient to optimize only on all $\overline{W}$, with rows lying in the span of $\{\phi_{x_i^j}\}_{ij}$. Representer theorems further state, if one minimizes the loss with a Tikhonov regularization $\mathcal{L}\left(\left\{f(x_i^j)\right\}_{ij}\right) + \lambda\|W\|_H^2$, then the optimal $W$ must have $W^\perp = 0$ (Schölkopf et al., 2001). The result also holds for minimizing $\mathcal{L}$ over $W$ with orthogonality constraints (Esser et al., 2024). The bottom line is that it suffices to restrict the search space for equation 1 to

$$\mathcal{F} = \left\{ f(x) = \overline{W}\phi_x = \sum_{i \in [n], j \in [p]} \alpha_i^j \langle \phi_{x_i^j}, \phi_x \rangle = \sum_{i \in [n], j \in [p]} \alpha_i^j k(x_i^j, x), \ \{\alpha_i^j\}_{i \in [n], j \in [p]} \in \mathbb{R}^h \right\}$$

which is a finite-dimensional space in $\mathbb{R}^{h \times np}$. Rewriting $\overline{W}$ in terms of $\{\alpha_i^j\}_{i \in [n], j \in [p]}$ follows, since each row of $\overline{W}$ lies in the span of $\{\phi_{x_i^j}\}_{ij}$. We categorize popular losses for self/un-supervised representation learning into: 1. reconstruction-based ($p = 1$), 2. contrastive, 3. joint embedding, and 4. predictive self-distillation ($p \geq 2$). The followings elaborate on these categories.

**Joint Embedding Objectives** $\mathcal{L}(Z_A, Z_B)$, the objective is to align embeddings $Z_A$ and $Z_B$, obtained from two views of the same data sample. We implement the kernelized version of two fa-

mous losses of this categories; Barlow Twins (BT) which minimizes redundancy between dimensions of learned embeddings by regularizing their cross-correlation matrix, and VICReg (Variance-Invariance-Covariance Regularization) which is designed to balance the variance, invariance, and covariance between the embeddings.

**Predictive Self-distillation Objectives** We implement kernel BYOL (Grill et al., 2020) loss function with the same definition.

**Contrastive Objectives** $\{x^p\} \equiv \{x, x^+, x^-\}$, denoting the anchor, the positive and the negative examples, respectively. We use the same definition as Esser et al. (2024) for the simple and spectral contrastive loss plus the aforementioned Tikhonov regularization term as the introduced orthogonality constraint. For SimCLR loss function we used the same exact definition as Chen et al. (2020).

**Reconstruction-based Objectives** KPCA generalizes PCA to non-linear feature space by mapping into RKHS $\mathcal{H}$ via feature map $\Phi$. Principal components in $\mathcal{H}$ are identified by maximizing the variance of the projected data while minimizing reconstruction loss. Kernel Auto-Encoder (KAE), minimizes a reconstruction loss with RKHS regularization as defined in Esser et al. (2024).

## 3 NYSTRÖM-BASED KERNEL REPRESENTATION LEARNING AT SCALE

**Motivation for Nyström Approximation.** As stated in Rudi et al. (2017), kernel ridge regression (KRR) with optimal statistical guarantees typically incur $O(n^2)$ time and memory costs. However, up to logarithmic factors, the same statistical accuracy can be achieved with $O(n\sqrt{n})$ time and $O(n)$ memory, using Nyström approximation to scale kernel methods.

The Nyström approximation used in Rudi et al. (2017); Abedsoltan et al. (2023) further restricts the above search space $\mathcal{F}$ based on the notion of a general kernel model. General kernel models correspond to functions $f : \mathcal{X} \to \mathbb{R}^h$ of the form $f(x) = \sum_{s=1}^{m} \beta_s k(c_s, x)$, where $c_1, \ldots, c_m \in \mathcal{X}$ are $m$ prespecified centers, and $\beta_1, \ldots, \beta_m \in \mathbb{R}^h$ are learnable parameters. In the kernel models discussed above, the centers correspond to the entire training dataset (all $np$ samples $x_i^j$ in our case). In the case of Nyström approximation for kernel regression, only a subset of the training data, usually called landmarks, is used as centers with the sample size $m \ll n$. Rudi et al. (2017) show that $m = O(\sqrt{n})$ landmarks suffice for optimal statistical accuracy, when they are sampled from training dataset, either uniformly or based on leverage scores.

**Nyström Approximation in Representation learning.** In the present context, where the training data consists of augmented samples, it is more natural to use $m \ll n$ tuples $(x_i^1, \ldots, x_i^p)$ as landmarks. We denote the subset of data selected as landmarks by $(\tilde{x}_i^1, \ldots, \tilde{x}_i^p)$, $i \in [m]$. Consequently, we minimize the loss in equation 1 over the class of functions

$$\mathcal{F} = \left\{ f(x) = \sum_{i \in [m], j \in [p]} \tilde{\alpha}_i^j k(\tilde{x}_i^j, x) + \gamma, \ \{\tilde{\alpha}_i^j\}_{i \in [m], j \in [p]} \in \mathbb{R}^h, \gamma \in \mathbb{R}^h \right\}$$

which is a subset of $\mathbb{R}^{mp \times h}$. The utility of the above function class is that one can numerically solve equation 1 for any arbitrary loss $\mathcal{L}$ using any available gradient descent-based solver, where the gradients are computed with respect to the terms $\{\tilde{\alpha}_i^j\}_{i \in [m], j \in [p]} \in \mathbb{R}^h$, a total of $m \times p \times h$ learnable weights, and $\gamma \in \mathbb{R}^h$ a learnable bias vector (ensuring that the model can capture global shifts in the embedding space which cannot be expressed as kernel combinations of the landmarks alone, which is important in self-supervised settings where embeddings are often centered implicitly by the loss function). Hence, the large-scale kernel representation learning framework, can be adapted to any un-/self-supervised loss function.

**Notation** $k_x \in \mathbb{R}^{mp}$ implies the vector of all kernel evaluations between $x$ and all landmarks, that is, $k(\tilde{x}_i^j, x)$ for $i \in [m], j \in [p]$. We denote $f \in \mathcal{F}$ as $f(x) = \tilde{A}^\top k_x + \gamma$, where the learnable term $\tilde{A} \in \mathbb{R}^{mp \times h}$ corresponds to a matrix with $\tilde{\alpha}_i^j$ as respective rows, and $\mathrm{Tr}(\cdot)$ computes the trace of a matrix, and $\odot$ indicates the element-wise (Hadamard) product. $K_{mm} \in \mathbb{R}^{mp \times mp}$ denotes the kernel matrix computed among all $m \times p$ augmented views in the landmark tuples, while $K_{mm}^j \in \mathbb{R}^{m \times m}$ is the kernel matrix computed among the $j$-th augmented views of all landmarks. Similarly, $K_{nm} \in \mathbb{R}^{np \times mp}$ refers to the kernel matrix computed between all training samples and all landmarks, while $K_{nm}^j \in \mathbb{R}^{n \times m}$ is the corresponding matrix only for $j$-th augmented

views. Some losses impose Tikhonov regularization $\|W\|_{\mathcal{H}}^2$ with $\mathrm{Tr}(\tilde{A}^\top K_{mm}\tilde{A})$ or the orthogonality constraint $WW^\top = I$ with $\tilde{A}^\top K_{mm}\tilde{A} = I$, restricting $\mathcal{F}$ to enforces $W$ to lie in the span of $\{\phi_{\tilde{x}_i^j}\}_{i\in[m],j\in[p]}$. We illustrate a two examples of how Nyström approximation integrates into kernelized self-supervised and unsupervised objectives:

**1)BT.** Using kernels, the embeddings are defined as $Z_A = K_A\tilde{A} + \gamma_A$ and $Z_B = K_B\tilde{A} + \gamma_B$, where $K_A$ (resp. $K_B$) are kernels between anchor (resp. positive) samples and their corresponding landmarks. The BT loss $\mathcal{L}_{\mathrm{BT}} = \sum_i(1 - C_{ii})^2 + \lambda\sum_{i\neq j}C_{ij}^2$ minimizes redundancy via the cross-correlation matrix $C$ defined element-wise as $C_{ij} = \dfrac{\sum_{b=1}^n z_{b,i}^A z_{b,j}^B}{\sqrt{\sum_{b=1}^n (z_{b,i}^A)^2}\sqrt{\sum_{b=1}^n (z_{b,j}^B)^2}}$, where in the kernel setting corresponding to the *i-th* and *j-th* dimension of the learned embeddings $z_{b,i}^A = \sum_{l=1}^m \tilde{A}_{il}k_A(x_l,x_b) + \gamma_i$, $z_{b,j}^B = \sum_{l'=1}^m \tilde{A}_{jl'}k_B(x_{l'},x_b) + \gamma_j$, $l$ indexes landmarks from anchor samples and $l'$ those from positive views.

**2)KPCA.** The kernelized reconstruction loss with Nyström approximation becomes

$$\frac{1}{n}\sum_{i=1}^n \left\|\Phi(x_i) - WW^\top\Phi(x_i)\right\|_{\mathcal{H}}^2 + \lambda\|W\|_{\mathcal{H}}^2$$

$$= \frac{\mathrm{Tr}(K)}{n} - \frac{1}{n}\mathrm{Tr}\left(\tilde{A}\tilde{A}^\top K_{nm}^\top K_{nm}\right) + \frac{1}{n}\mathrm{Tr}\left(\tilde{A}\tilde{A}^\top K_{mm}\tilde{A}\tilde{A}^\top K_{nm}^\top K_{nm}\right) + \lambda\,\mathrm{Tr}\left(\tilde{A}^\top K_{mm}\tilde{A}\right).$$

**From Analytic to Algorithmic Design.** A key distinction between our setting and classical KRR analyses (Rudi et al., 2017; Vecchia et al., 2024; Li et al., 2016) is that the latter admits closed-form solutions derived from the analytic form of the loss. In contrast, we formulate the kernel SSL objectives to be optimizable via first-order gradient-based methods such as Adam, (Kingma & Ba, 2014). This shift necessitates a different design philosophy, developed in the following subsections, to make scalable kernel representation learning tractable. Our framework is depicted in algorithm 1.

### 3.1 PRINCIPAL COMPONENT INITIALIZATION

To improve optimization, we propose a principled parameter initialization scheme for $\tilde{A}$, rather than using random initialization. We begin with the standard Nyström approximation $K_{nn} \approx K_{nm}K_{mm}^\dagger K_{mn}$, (Williams & Seeger, 2000) where $K_{mm}^\dagger$ denotes the Moore–Penrose pseudoinverse of $K_{mm}$. Let the rank $h$ truncated eigendecomposition of $K_{mm}$ be $K_{mm} \approx U_h\Lambda_h U_h^\top$, with $U_h \in \mathbb{R}^{m\times h}$ the top h eigenvectors and $\Lambda_h \in \mathbb{R}^{h\times h}$ the corresponding eigenvalues, the pseudoinverse is then $K_{mm}^\dagger \approx U_h\Lambda_h^{-1}U_h^\top$. Substituting this into the Nyström approximation and defining $\Phi = K_{nm}U_h\Lambda_h^{-1/2} \in \mathbb{R}^{n\times h}$, yields the following form for the low-rank kernel approximation

$$K_{nn} \approx (K_{nm}U_h\Lambda_h^{-1/2})(K_{nm}U_h\Lambda_h^{-1/2})^\top = \Phi\Phi^\top \tag{2}$$

Now consider our parameter matrix $\tilde{A} \in \mathbb{R}^{m\times h}$ as a projection from the kernel space into an h-dimensional feature space, such that $f(\mathcal{X}) = K_{nm}\tilde{A} \in \mathbb{R}^{n\times h}$; comparing with the Nyström feature map $\Phi$, we observe that a natural initialization is $\tilde{A}_0 = U_h\Lambda_h^{-1/2}$. Numerical results presented in E show our method outperforms random initialization significantly.

### 3.2 IMPROVING OPTIMIZATION VIA SECOND ORDER METHODS

We minimize equation 1 using iterative first-order methods, which update parameters as $\tilde{A}_{t+1} = \tilde{A}_t - \eta_t\nabla f(\tilde{A}_t)$. While effective in practice, such methods disregard curvature information, motivating second-order approaches, which exploit the Hessian to rescale gradients according to local curvature, yielding Newton-like updates. In neural networks, however, the Hessian is impractical: (i) computing and inverting it is prohibitively costly, and (ii) it is indefinite, with negative eigenvalues reflecting non-convex regions. The Fisher Information Matrix (FIM) is often used as a Positive Semi-Definite (PSD) surrogate, but remains expensive, motivating diagonal, block-diagonal, or low-rank approximations (Becker & Cun, 1989; Schaul et al., 2013; George et al., 2018; Frantar et al.,

2021). For kernel SSL, these approaches face two issues: they are tailored to likelihood-based losses, and diagonal approximations discard essential cross-dimensional correlations. Since kernel matrices explicitly encode covariances, off-diagonal curvature is crucial for representation learning. Fortunately, our model's modest parameterization makes it tractable to exploit Hessian structure, preserving this information.

**General Loss Preconditioner** Preconditioned gradient descent is defined as $\tilde{A}_{t+1} = \tilde{A}_t - \eta_t P^{-1} \nabla f(\theta_t)$, where $P$ is the preconditioner. Rudi et al. (2017) introduce a preconditioner of the form $(\frac{n}{m} K_{mm}^2 + \lambda n K_{mm})$, which approximates the Hessian in KRR. Empirically, we find that the component $P = K_{mm} + \lambda I$, which models the geometry of the input kernel space, improves optimization for several loss functions. We propose this as a general loss preconditioner. See the algorithm in appendix D.

**Generalized Gauss–Newton (GGN) Hessian Approximation** GGN approximation (Korbit et al., 2024) replaces the Hessian with a PSD surrogate of the form $H_{\mathrm{GN}} = J^\top Q J$, constructed using only first-order sensitivities. This provides curvature information without requiring second-order derivatives and is applicable to both regression (MSE-type losses) and multi-class classification (cross-entropy losses). In our setting, we treat the BT objective as a nonlinear least-squares problem (analogous to MSE) and the SimCLR objective as a softmax cross-entropy. In the following, we derive the corresponding GGN approximations for each loss. More details in appendix C.

**1) BT.** BT loss $L = \sum_{i=1}^{k} (C_{ii} - 1)^2 + \lambda_{reg} \sum_{i \neq j} C_{ij}^2$, can be thought of as a sum of squared residuals $L = \|r\|_2^2$. Hence, we reframe it into a non-linear least squares problem $L = \|W \odot (C - I)\|_F^2$, $W \in \mathbb{R}^{h \times h}$, $W_{ii} = 1$, $W_{ij} = \sqrt{\lambda_{reg}}$. We then approximate $\nabla^2 L$ as $H_{GN} = J^T J$ ( where $J = \frac{\partial r}{\partial \mathrm{vec}(A)}$). Our goal is to have a preconditioner that solves the systems $H_{GN} \cdot p = g$ (g is the gradient and p the preconditioned gradient). The Conjugate Gradient (CG) solver does this without forming $H_{GN}$ explicitly, only requiring a function that can compute the Hessian-Vector Product (HVP) as $H_{GN} \cdot d = (J^T J) \cdot d = J^T (J \cdot d)$ for an arbitrary direction vector $d$, via a 2-step process; (1) a Jacobian-Vector Product (JVP):$u = J \cdot d$, (2) a Vector-Jacobian Product (VJP): $J^T u$.

**2) SimCLR.** Considering SimCLR loss as a composition of the parameter-to-logits map and the softmax cross-entropy loss on each logits row, let f(A) denote the vectorized logits and $J = \frac{\partial f}{\partial \mathrm{vec}(A)}$. The gradient and the GN Hessian are $g = J^\top r$, $H_{\mathrm{GN}} = J^\top Q J$, where $r_i = p_i - e_{y_i} \in \mathbb{R}^C (i = 1, \ldots, C)$ (when $\mathbf{r} = \mathrm{softmax}(\mathrm{logits}) - \mathbf{y}_{\mathrm{one\text{-}hot}}$), are the pseudo-residuals and Q is block-diagonal with per-row blocks $Q_i = \mathrm{diag}(p_i) - p_i p_i^\top$ and $p_i = \sigma(Z_{i,:}) \in \mathbb{R}^C$ are per-row softmax probability vectors. Our preconditioner solves the regularized linear system $(J^\top Q J + \lambda I)s = g$, using CG without forming the $H_{\mathrm{GN}}$. The required Hessian-vector product is computed via the 2-step JVP–$Q$–VJP procedure: $\mathrm{HVP}(d) = J^\top (Q(Jd)) + \lambda d$, where $Q(Jd)$ is evaluated per-row by $Q_i v = p_i \odot v - p_i (p_i^\top v)$, which costs $O(C)$ per row and does not require forming $Q_i$ explicitly.

### 3.3 Efficient Landmark Selection

Since Nyström is a low-rank approximation, the landmarks define the column space of $K_{nm}$ and the subspace for $\tilde{A}_0$ ( 3.1); hence, selecting them carefully ensures the initial projection captures the most informative directions in kernel space, improving convergence and representation quality.

We choose the Nyström landmarks using two approaches; first one is the kmeans++ seeding algorithm (Arthur & Vassilvitskii, 2007) in which we choose the first landmark $c_1 \sim \mathrm{Uniform}(\mathcal{X})$, and initialize the set of selected landmarks as $C = \{c_1\}$, then we continue adding to this list iteratively with the probability $P(x_j) = \frac{D(x_j)^2}{\sum_{k=1}^{n} D(x_k)^2}$, where $D(x_j)^2 = \min_{c \in C} \|x_j - c\|_2^2$ until $|C| = m$. Secondly, we propose a randomized diagonal estimation of the q-approximate leverage scores leveraging the Hutchinson's estimator (Hutchinson, 1989). Let $K \in \mathbb{R}^{n \times n}$ be the kernel matrix, where $K_{ij} = \kappa(x_i, x_j)$. Leverage score for the $j$-th point is defined as $l_j(\lambda) = (K(K + \lambda n I)^{-1})_{jj}$ (Rudi et al., 2017; 2018). Since directly computing this inverse would require $O(n^3)$ operations and $O(n^2)$ memory, we propose a randomized algorithm using Hutchinson's estimator to approximate the diagonal of $M = K(K + \lambda n I)^{-1}$. We generate a random sketching matrix $\Pi \in \mathbb{R}^{n \times s}$, (Halko et al., 2011) where each entry $\Pi_{ij}$ is drawn independently from a distribution (e.g., a standard normal or a

sparse Rademacher distribution). We define $Z = (K + \lambda nI)^{-1}\Pi$, $M\Pi = KZ$ then we solve each linear system $(K + \lambda nI)z_j = p_j, \forall j$, via Conjugate Gradients (Hestenes & Stiefel, 1952). As a result, we have the vector of approximated leverage scores $\hat{\ell}(\lambda) = \sum_{\text{cols}}\big(\Pi \odot (M\Pi)\big)$, and choose landmarks with the probability $P(x_j) = \dfrac{\hat{\ell}_j(\lambda)}{\sum_{k=1}^{n}\hat{\ell}_k(\lambda)}$. Related results can be found in appendix F.

**Kernel Choice** KREPES supports various kernels. Among these, neural kernels have gained particular prominence, with empirical Neural Tangent Kernels (eNTKs) proving effective in practice. Details about our scalable Nyström eNTK implementation can be found in appendix A.

---

**Algorithm 1** KREPES: Kernel Representation Learning at Scalable via Nyström Approximation

---

**Require:** Unlabeled data $\mathcal{X}$, number of landmarks $m$, Output dimension $h$, Batch size $B$, Learning rate $\eta$, Preconditioner type $\mathcal{P} \in \{\text{None, General, GGN}\}$.
**Ensure:** Learned projection $\tilde{A} \in \mathbb{R}^{m \times h}$, bias $\gamma \in \mathbb{R}^h$.
    **1: Nyström Initialization**
  1: landmarks $\mathcal{Z} = \{\tilde{x}_1, \ldots, \tilde{x}_m\} \subset \mathcal{X}$ via K-Means++ or Leverage Score Sampling     ▷ Sec. 3.3
  2: $K_{mm} \approx U_h \Lambda_h U_h^\top$, $\tilde{A}^{(0)} \leftarrow U_h \Lambda_h^{-1/2}$ and $\gamma^{(0)} \leftarrow \mathbf{0}$.             ▷ Sec. 3.1
    **2: Optimization**
  3: **for** epoch $t = 1 \ldots T$ **do**
  4:     **for** batch $X \subset \mathcal{X}$ **do**
  5:         $K_{nm}^A = k(X_A, \mathcal{Z})$ and $K_{nm}^B = k(X_B, \mathcal{Z})$ for augmented views $X_A, X_B$
  6:         **Forward Pass:**
  7:         $Z_A \leftarrow K_{nm}^A \tilde{A} + \gamma, \quad Z_B \leftarrow K_{nm}^B \tilde{A} + \gamma.$
  8:         Compute $\mathcal{L}(Z_A, Z_B)$
  9:         **Backward Pass:**
10:         $g \leftarrow \nabla_{\tilde{A}}\mathcal{L}.$
11:         **if** $\mathcal{P} ==$ None **then**
12:             $p \leftarrow g$
13:         **else if** $\mathcal{P} ==$ General **then**
14:             Construct $P \approx K_{mm} + \lambda I$.
15:             Solve $Pp = g$ (via Conjugate Gradient (CG)).
16:         **else if** $\mathcal{P} ==$ GGN **then**
17:             $\text{HVP}(v) = \text{vjp}(\text{res}, \tilde{A}, \text{jvp}(\text{res}, \tilde{A}, v)) + \lambda v.$
18:             Solve $\mathbf{H}_{GN}p = g$ (via CG).
19:                                   ▷ Appendix C for derivations per loss
20:         **end if**
21:         **Update:** $\tilde{A} \leftarrow \tilde{A} - \eta p, \quad \gamma \leftarrow \gamma - \eta \nabla_\gamma \mathcal{L}.$
22:     **end for**
23: **end for**
24: **return** $\tilde{A}, \gamma$

---

## 4 Scalability Analysis

Table 1 reports downstream accuracy and training time for KREPES with eNTK and the corresponding neural network (NN), evaluated across four SSL losses and four large datasets. Datasets (size) include an expanded variant of Adult (1M), Higgs (1M), CoverType (581k), CIFAR-10 (50k). The results show that (1) our method performs comparable to the corresponding NN, and (2) the kernel-based framework scales efficiently to large datasets. A linear classifier (Fan et al., 2008) is used on top of the pretrained SSL features. Further implementation details, data and the neural networks specifications are in appendix B. NNs are generally faster to train, however KREPES additionally provides interpretability which we discuss in next section. In appendix G we plot the comparison of the performance and training time of the full kernel vs. KREPES on different subset sizes of the Adult dataset. We observe that KREPESpreserves the predictive power of the full kernel, despite operating at substantially reduced time complexity.

**Effect of Preconditioning** Table 2 reports results without any preconditioner and with two preconditioning setups for BT and SimCLR across multiple datasets. Incorporating preconditioners

Table 1: Scalability results in downstream test accuracy with time in parenthesis (as average time in 1 forward and backward pass times number of epochs needed for convergence) in seconds, for KREPES with eNTK and NN. The number of training learnable parameters in KREPES for the 4 datasets are between 500k-900k, whereas the #parameters in NN is 5M (tabular), and 22M (vision).

| Dataset | BT | | SimCLR | | VICReg | | BYOL | |
|---|---|---|---|---|---|---|---|---|
| (#samples) | KREPES | NN | KREPES | NN | KREPES | NN | KREPES | NN |
| Adult | 83.78 | 83.72 | 83.67 | 83.79 | 83.77 | 82.65 | 83.76 | 83.50 |
| (1M) | (0.048) | (0.021) | (0.372) | (0.036) | (0.060) | (0.022) | (0.545) | (0.024) |
| Higgs | 66.66 | 66.69 | 68.50 | 69.60 | 67.70 | 56.47 | 66.24 | 67.47 |
| (1M) | (10.00) | (0.112) | (1.463) | (0.312) | (0.240) | (0.015) | (0.066) | (0.017) |
| CoverType | 71.32 | 73.71 | 75.17 | 68.10 | 73.85 | 71.08 | 70.31 | 72.36 |
| (581k) | (11.05) | (0.021) | (20.19) | (0.019) | (22.10) | (2.640) | (0.070) | (0.240) |
| CIFAR-10 | 90.31 | 85.32 | 89.54 | 84.82 | 89.88 | 84.52 | 88.76 | 78.12 |
| (50k) | (6.701) | (1.119) | (0.625) | (0.321) | (24.60) | (0.671) | (0.098) | (0.428) |

Table 2: Preconditioning results in downstream test accuracy for BT and SimCLR across datasets, with preconditioning types (No Preconditioner, General Loss, GGN Hessian Approximation).

| Dataset | None | | General | | GGN | |
|---|---|---|---|---|---|---|
| | BT | SimCLR | BT | SimCLR | BT | SimCLR |
| Adult-1M | 83.58 | 81.67 | 83.55 | 82.10 | 83.78 | 83.67 |
| CoverType | 68.83 | 52.67 | 71.32 | 62.12 | 70.93 | 75.17 |
| CIFAR-10 | 89.11 | 89.17 | 88.56 | 89.48 | 90.31 | 89.54 |

Figure 1: Eigenvalue spectrum comparison of $\tilde{A}^\top \tilde{A}$ under different preconditioners with BT.

consistently improves performance, with loss-specific GGN Hessian approximation providing the largest gains. This advantage stems from their ability to exploit second-order information tailored to the underlying SSL loss, in contrast to general preconditioners. Moreover, figure 1 shows the eigenvalue spectra of $\tilde{A}^\top \tilde{A}$ for different preconditioning setups. Preconditioners generally flatten the spectrum, retaining more significant eigenvalues and thus increasing the effective rank of the representation. This indicates that preconditioning enables learning along more informative directions, which correlates with improved downstream performance. While the general preconditioner behaves differently across datasets, the loss-specific variant consistently preserves a rich spectrum.

## 5 INTERPRETABILITY ANALYSIS

Our goal is to obtain *interpretable representations* — that is, embeddings such that one can explain why a test sample $x_t$ is mapped to a particular location in the representation space. One way to achieve this, is to trace or attribute the influence of training data or concepts to individual predictions in an understandable way. Yeh et al. (2018) leverage representer theorem to explain neural network predictions, and for a test point $x_t$ write output as $\Phi(x_t, \Theta^*) = \sum_{i=1}^n k(x_t, x_i, \alpha_i)$, where $k(x_t, x_i, \alpha_i) = \alpha_i f_i^\top f_t$ is the representer value of training point $x_i$ for $x_t$, and $f_t$ denotes the pre-activation of $x_t$. The coefficients $\alpha_i$ are derived from the training loss, making $x_i$ a representer point influencing the prediction.

Building on this intuition, since we propose a kernel representation learning framework, representer point interpretation is inherent to the nature of KREPES, meaning we don't need to derive the coefficients $\alpha_i$, they're simply our learned parameters. Further, unlike Yeh et al. (2018), and Engel et al. (2024), we provide interpretation of the self-supervised learned representations rather than explaination of the supervised predictions. Moreover, instead of using training samples as representer points we take advantage of our Nyström anchors. After training, the learned representations take the form $Z = K_{nm}\tilde{A} + \gamma$; we compute row norms of $\tilde{A}$ as $\omega_l = \|\tilde{a}_l\|_2$. Let $\pi$ be a permutation of the landmark indices such that $\omega_{\pi(1)} \geq \omega_{\pi(2)} \geq \cdots \geq \omega_{\pi(m)}$. We then identify the minimum number of top-ranked landmarks required to represent all $C$ classes in the dataset. Let $Y$ be the set of all class labels and $y(x_l)$ be the ground-truth label of landmark $x_l$. We define the class coverage metric as: $\kappa = \min\left\{k \in \{1,\ldots,m\} \;\middle|\; Y \subseteq \{y(x_{\pi(i)})\}_{i=1}^k\right\}$. Table 3 reports $K$ alongside downstream classification test accuracy. We observe that smaller $K$ values align with higher downstream accuracy, indicating that the representations allocate high-norm landmarks across diverse categories. Remarkably, despite the absence of label supervision, the SSL models prioritize landmarks that span distinct semantic classes, reflecting an implicit alignment with downstream performance. Our interpretability framework in depicted in algorithm 2.

## 5.1 SAMPLE SPECIFIC INFLUENCE SCORE

Inspired by Tsai et al. (2023) we define the influence of landmark $x_l$ on a sample's representation as the product of two factors: (i) local importance; how similar is the sample to the landmark $k(x_{test}, x_l)$ (ii) global importance; how important is the landmark to the overall model captured by the norm of the landmark's corresponding column in the learned projection matrix $\|\tilde{A}_l\|$ as $\iota_{l\to t} = \underbrace{k(x_{test}, x_l)}_{\text{local importance}} \cdot \underbrace{\|\tilde{A}_l\|}_{\text{global importance}}$ . This is in line with what's referred to as global vs. local interpretability in Doshi-Velez & Kim (2017). Figure 2 shows the corresponding results.

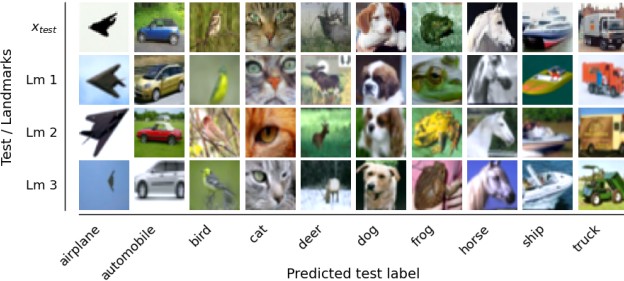

Figure 2: Sample-specific influential landmarks. First row, $x_{test}$, then top-3 landmarks, ranked by their influence score.

Table 3: Minimum k for top-k landmarks covering all 10 CIFAR-10 classes across SSL losses, with corresponding test accuracy (acc).

| Loss Function | acc(%) | $\kappa$ |
|---|---|---|
| BT | 90.31 | 12 |
| VICReg | 89.88 | 18 |
| SimCLR | 89.54 | 25 |
| BYOL | 88.76 | 27 |
| Spectral Cont. | 87.95 | 81 |

## 5.2 CONCEPTUAL INFLUENCE PROFILE

We next interpret representations through a concept-alignment lens. Having identified most influential landmarks via the representer framework as an immediate benefit of KREPES we quantify how strongly each aligns with a given concept. This allows us to attribute a sample's prediction to specific concept-aligned landmarks. Inspired by Kim et al. (2018) on Concept Activation Vectors (CAV), we aim to answer the interpretability question: *How much did a specific concept c, as embodied by landmark l, contribute to the prediction for test sample $x_{test}$?*

For each test sample, we generate a profile of how a concept influences its prediction by defining

$$\text{Score}(l \to t, c) = \underbrace{(Z_m[l, :] \cdot v_c)}_{\text{Concept Alignment in Landmark}} \times \underbrace{(k(x_{test}, c_j) \cdot \|A_l\|)}_{\text{Influence Score} := \iota_{l \to t}} \quad (3)$$

r where $v_c$ is the CAV for concept $c$, $Z_m \in \mathbb{R}^{m \times k}$ is the representation of all the landmarks. Further, we can aggregate 3 by concept yielding the total influence of a concept c on the prediction for $x_{test}$ as the sum of the scores from top-$N$ landmarks: $\Psi_N(t, c) = \sum_{l=1}^{N} \text{Score}(l \to t, c)$. For example,

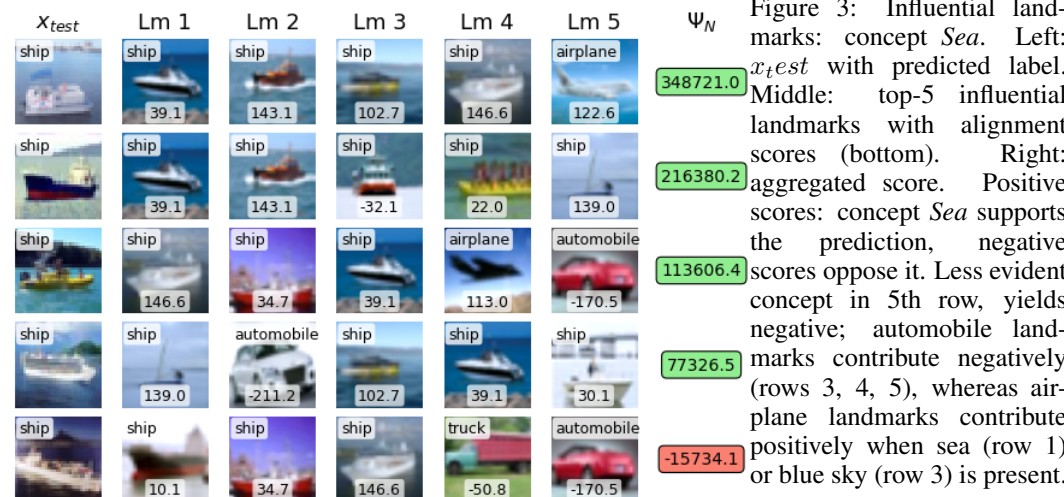

Figure 3: Influential landmarks: concept *Sea*. Left: $x_test$ with predicted label. Middle: top-5 influential landmarks with alignment scores (bottom). Right: aggregated score. Positive scores: concept *Sea* supports the prediction, negative scores oppose it. Less evident concept in 5th row, yields negative; automobile landmarks contribute negatively (rows 3, 4, 5), whereas airplane landmarks contribute positively when sea (row 1) or blue sky (row 3) is present.

the top-5 influential landmarks often exhibit strong alignment with the concept relevant to the test sample, highlighting which landmarks drive the prediction. Figure 3 illustrates this for the concept Sea (from CIFAR-100). Each landmark's alignment score is shown at the bottom center, and the rightmost column reports the aggregated score $\Psi_N(t, c)$ with $N = 5$. A positive score indicates that concept Sea supports the prediction, while a negative score reflects opposing influence. For instance, in the last row, the concept Sea is less evident in the test image, yielding a negative score as the model is trained on clearer examples of concept Sea. Landmarks from classes such as automobile contribute negatively (rows 3, 4, and 5), whereas airplane landmarks can contribute positively when there is sea in the image (row 1), or very blue sky (row 3). Details in the appendix I.

---

**Algorithm 2** KREPES Interpretability: Influence Scores and Conceptual Profiles

---

**Require:** $x_{\text{test}}$, $\mathcal{Z} = \{\tilde{x}_l\}_{l=1}^m$, Learned $\tilde{A}$, Concept $c$ with Concept Activation Vector $v_c$.
**Ensure:** Top-$N$ influential landmarks, Conceptual Score $\Psi$.
    **Step 1: Influence Score**
1: **for** landmark $l = 1 \ldots m$ **do**
2:     $I_{\text{local}}(l) \leftarrow k(x_{\text{test}}, \tilde{x}_l), \quad I_{\text{global}}(l) \leftarrow \|\tilde{A}_{l,:}\|_2, \quad S(l \rightarrow x_{\text{test}}) \leftarrow I_{\text{local}}(l) \times I_{\text{global}}(l)$
3: **end for**
4: Identify Top-$N$ landmarks: $\zeta_{\text{top}} \leftarrow \text{argsort}(S(l \rightarrow x_{\text{test}}))[: N]$.
5: **return** $\zeta_{\text{top}}$
    **Step 2: Conceptual Profile**
6: $\Psi(x_{\text{test}}, c) \leftarrow 0$
7: **for** $l \in \zeta_{\text{top}}$ **do**
8:     $z_l \leftarrow k(\tilde{x}_l, \mathcal{Z})\tilde{A}, \quad \psi_l \leftarrow \langle z_l, v_c \rangle$
9:     $\Psi(x_{\text{test}}, c) \leftarrow \Psi(x_{\text{test}}, c) + S(l \rightarrow x_{\text{test}}) \cdot \psi_l$
10: **end for**
11: **return** $\Psi(x_{\text{test}}, c)$         ▷ Positive $\Psi$ implies concept supports prediction

---

## 6 CONCLUSION

We introduced KREPES, a unified and scalable framework for kernel-based self-supervised representation learning. By combining Nyström approximation with gradient-based optimization, KREPES supports diverse loss functions while remaining efficient at scale. Our experiments demonstrate competitive performance on large benchmarks, while enabling principled interpretability through representer landmarks and a proposed conceptual influence profile.

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

## A EFFICIENT EMPIRICAL NTK CALCULATION

KREPES supports various kernels, including RBF, Laplacian, and user-defined kernels. We observed empirically that empirical Neural Tangent Kernels (eNTKs) by far have better performance.

We adopt eNTKs instantiated from a wide range of architectures, including MLPs, CNNs with pooling, ResNets (18, 34, 50), and self-attention blocks.

For a network with parameters $\theta \in \mathbb{R}^P$ and $C$ output classes, let $f(x; \theta) \in \mathbb{R}^C$, $\phi_{\text{eNTK}}(x) = \frac{\partial f(x;\theta)}{\partial \theta} \in \mathbb{R}^{P \times C}$. The eNTK between inputs $x_i, x_j$ is defined by the Jacobian inner product $K_{\text{eNTK}}(x_i, x_j) = \phi_{\text{eNTK}}(x_i)^\top \phi_{\text{eNTK}}(x_j) \in \mathbb{R}^{(N \cdot C) \times (N \cdot C)}$. When the output layer is randomly initialized, the expected kernel factorizes as $\mathbb{E}[K_{\text{eNTK}}] \approx I_C \otimes K_0$, ($I_C := C \times C$ identity, $K_0 \in \mathbb{R}^{N \times N}$ is the kernel for a single logit)(Lee et al., 2020). In practice, we approximate $K_{\text{eNTK}} \approx I_C \otimes K$, where $K$ is computed from one randomly initialized output head. To further accelerate computation, we adapt the parallel eNTK method of Wei et al. (2022) and integrate Nyström approximation.

## B DOWNSTREAM CLASSIFICATION ACCURACY

For performance evaluation, we report downstream classification test accuracy. Unlike conventional linear probing in self-supervised learning, we do not use the entire labeled training set. Instead, we first train KREPES in a fully self-/unsupervised manner without labels, and then train a linear classifier on representations of only 10% of the labeled data, using labels solely for the classifier training.

In table 1, the reported times are measured between a timestamp recorded before `optimizer.zero-grad` and another taken after either `loss.backward` or, when a preconditioner is used, `preconditioner.step`.

### B.1 DATA PRE-PROCESSING AND AUGMENTATION

For all datasets, we ensure proper feature normalization. In tabular datasets, categorical columns are encoded numerically, while numerical features are standardized. To enhance generalization, we apply data augmentation by adding Gaussian noise and randomly dropping features.

For CIFAR-10 we normalize using dataset's mean and standard variation. For augmentation we use the following data augmentation pipeline implemented with `torchvision.transforms`:

```
augmentation = transforms.Compose([
    transforms.ToPILImage(),
    transforms.RandomResizedCrop(size=(224, 224), scale=(0.5, 1.0)),
    transforms.RandomHorizontalFlip(p=0.5),
    transforms.ToTensor(),
    transforms.Normalize(mean=[0.485, 0.456, 0.406],
                         std=[0.229, 0.224, 0.225]),
])
```

### B.2 BASELINE ARCHITECTURE AND THE EMPIRICAL NTK

In all the comparisons with the neural networks, we set the neural network of the corresponding eNTK kernel with which KREPES is trained as the baseline. It's important to note that all the parameters in the neural network are trained, however in KREPESwe only train the matrix $\tilde{A}$, and the bias vector $\gamma$. For cifar10, as the kernel, we use the eNTK of a ResNet34 with pre-trained on ImageNet weights; therefore, as the baseline, we use the pre-trained (on ImageNet dataset) ResNet34. For CIFAR-10 KREPES has $(m \times h)$ $300k/500k$ parameters $m$ being the number of landmarks setting to 2000 and $h$ depending on the loss function being $150/250$.

For all the tabular datasets we use the following architecture as the neural network and the eNTK of that as our kernel in KREPES:

**Description.** We use a residual MLP backbone with integrated self-attention, which we refer to as *ResMLP*. The input $x \in \mathbb{R}^d$ is first projected to width $w$ using a linear layer followed by GELU activation. A residual self-attention block (LayerNorm + single-head self-attention + skip connection) is then applied. The model further includes $B$ bottleneck MLP blocks, each consisting of a two-layer feedforward network with hidden dimension $w/r$ (where $r$ is the bottleneck factor), residual connection, and GELU activation. Finally, the representation is projected to the output dimension. Unless otherwise stated, we use $w = 1032$, $B = 3$, $r = 4$, and single-head attention in all experiments. The full PyTorch implementation is provided in the supplementary material.

---

**Algorithm 3** ResMLP Forward Pass

---

**Require:** Input $x \in \mathbb{R}^d$, width $w$, bottleneck factor $r$, number of blocks $B$
1: $x \leftarrow \text{Linear}(d \rightarrow w)(x)$
2: $x \leftarrow \text{GELU}(x)$
3: $x \leftarrow x + \text{SelfAttention}(\text{LayerNorm}(x))$
4: **for** $i = 1$ to $B$ **do**
5: $\quad h \leftarrow \text{Linear}(w \rightarrow w/r)(x)$
6: $\quad h \leftarrow \text{GELU}(h)$
7: $\quad h \leftarrow \text{Linear}(w/r \rightarrow w)(h)$
8: $\quad x \leftarrow \text{GELU}(x + h)$
9: **end for**
10: **return** $x$

---

The feature dimension d varies across datasets, influencing the total number of parameters: $d = 110$ for Adult, $d = 54$ for CoverType, and $d = 28$ for Higgs. Nonetheless, all resulting neural networks have roughly 5 million parameters. The total number of KREPES parameters ($m \times h$) depends on the number of landmarks and the representation dimension $h$. For all datasets except MNIST, we use 2000 landmarks, yielding between $500k$ and $900k$ KREPES parameters depending on h.

### B.3 HYPER-PARAMETER TUNING

For our experiments, we conduct hyper-parameter search mainly on loss function coefficients, and not general training parameters (except learning-rate and weight-decay in Adam optimizer) such as batch sizes. We use random search, which explores the hyper-parameter space by selecting values based on specified distributions. To this end, we employ the `Sweeps` framework provided by `Weights & Biases`. We conduct the same equal procedure for both KREPES and the neural networks.

**Hyper-parameter Tuning Pipeline** In the following, we describe the hyper-parameter tuning pipeline executed for each loss-dataset combination.

Training dataset is divided into an unlabeled training set and a labeled validation set. For image datasets, $2\%$ of the data is used for validation, while for tabular data sets, $10\%$ is used. For each hyper-parameter combination, we first learn a representation on the unlabeled training set, applying early stopping to prevent overfitting, and then train a linear classifier with only the small labeled subset. We refer to a single iteration of this procedure as a sweep. In general, we conduct only 3/4 sweeps per loss-dataset combination. We select the best hyper-parameter combination based on the accuracy of the downstream linear classifier. For imbalanced datasets, we report the `balanced_accuracy_score` instead. We acknowledge that selecting loss hyper-parameters based on labeled data compromises the self-supervised or unsupervised nature of the representation learning approach. However, to the best of our knowledge, no superior method currently exists for hyper-parameter selection. For a detailed discussion of this challenge, see Locatello et al. (2019).

**Hyper-parameter Ranges and Distributions** Next, we provide a detailed description of the hyper-parameters and their respective ranges. Unless stated otherwise, the parameters are sampled from a log-uniform distribution.

**General Training Hyper-parameters** The different loss functions are optimized using Adam and the cosine annealing learning rate scheduler from `torch.optim`, with a maximum of 50 iterations

and a minimum learning rate of $10^{-5}$ together with warm-up. The initial learning rate is determined through hyper-parameter tuning and sampled from the range $[10^{-4}, 10^{-1}]$. The batch size is always 256. For early stopping, the maximum number of epochs is between 60 and 80 with a patience of 10 to 20.

**Loss Function Hyper-parameters** For the simple contrastive loss, spectral contrastive loss, BT, and KPCA, the regularization coefficient $\lambda$ is sampled from the range $[10^{-5}, 100]$. For SimCLR, the temperature parameter $\tau$ is sampled from the interval $[10^{-3}, 100]$. For VICReg, the parameters $\tau$, $\lambda$, $\mu$, and $\nu$ are all sampled from the range $[10^{-5}, 100]$. Finally, for KAE, the regularization and covariance coefficients are also sampled from the interval $[10^{-5}, 100]$.

### B.4 DIFFERENT PERFORMANCE OF NN BASELINE AND SOTA ON CIFAR-10

Performance evaluation results in terms of downstream classification accuracy of a linear classifier on CIFAR-10, reported in Chen et al. (2020), Bardes et al. (2022), Grill et al. (2020) is 90%+, which is higher than the best performance of the ResNet-34 baseline reported in our work (85.32%). However, we note several crucial differences in our setup which we needed to incorporate in order to make the comparison with KREPES fair for the purpose of this paper:

- All of the mentioned SOTA works, use pretrained ResNet-50

- Linear classifier uses only 10% of labeled data; as explained in Appendix B, our evaluation protocol does not use the full labeled CIFAR-10 training set. After self-supervised training, the linear classifier is trained on only 10% of the labeled data. This differs significantly from standard linear probing protocols, which typically use all labels.

- No supervised fine-tuning; we strictly evaluate representations via linear classification without any supervised fine-tuning, whereas many SSL papers fine-tune or implicitly improve representations through heavy augmentations.

- Much lighter augmentation pipeline; our augmentation pipeline is intentionally simpler and less diverse (also depicted in details in appendix B) than those used in the above-mentioned works, which is known to affect absolute accuracy.

- No projection head for the neural network baseline; to ensure a fully fair comparison with KREPES, we also do not use any projection head in the neural network baseline. In contrast, all high-performing SSL works (including SimCLR, BYOL, Barlow Twins, and DINO) rely critically on multi-layer projection heads, which improve performance.

Equal conditions for KREPES and neural network baselines: most importantly, all of these constraints apply equally to KREPES and the neural network baselines. Thus, while the absolute accuracy is lower than typical ResNet-50 SSL results reported in the literature, the comparison between the two models under our protocol is fair and controlled.

## C DERIVATIONS OF GENERALIZED GAUSS-NEWTON PRECONDITIONERS

Let the kernel matrix be $K \in \mathbb{R}^{n \times m}$, and the learned parameter matrix be $\tilde{A} \in \mathbb{R}^{m \times h}$. The representations are given by the encoder function $f_\theta$

$$\mathbf{Z} = \mathbf{K}\tilde{\mathbf{A}} + \gamma \quad \in \mathbb{R}^{n \times h}$$

where $\theta = \text{vec}(\tilde{\mathbf{A}})$ denotes the flattened parameters. For the derivation, we assume $\gamma$ is absorbed or negligible for the curvature of $\tilde{\mathbf{A}}$. We define the Jacobian of the representations with respect to the parameters as $\mathbf{J}_Z = \frac{\partial \text{vec}(\mathbf{Z})}{\partial \theta}$.

### C.1 BARLOW TWINS (BT)

We treat the Barlow Twins objective as a Non-Linear Least Squares problem. This allows us to utilize the classic Gauss-Newton approximation, which guarantees a Positive Semi-Definite (PSD)

curvature matrix. The original loss is defined on the cross-correlation matrix $\mathcal{C} \in \mathbb{R}^{h \times h}$ between normalized batches $\hat{\mathbf{Z}}_A$ and $\hat{\mathbf{Z}}_B$

$$\mathcal{L}_{BT} = \sum_i (1 - \mathcal{C}_{ii})^2 + \lambda_{reg} \sum_{i \neq j} \mathcal{C}_{ij}^2$$

We formally reframe this as the squared $L_2$ norm of a **residual vector** $r(\theta)$. Let $\mathbf{W} \in \mathbb{R}^{h \times h}$ be a weighting matrix, where $W_{ii} = 1$ and $W_{ij} = \sqrt{\lambda_{reg}}$ for $i \neq j$. We define the residual map $r : \mathbb{R}^{m \times h} \to \mathbb{R}^{h^2}$ as

$$r(\theta) = \text{vec}\left(\mathbf{W} \odot (\mathcal{C}(\theta) - \mathbf{I})\right)$$

The loss becomes

$$\mathcal{L}_{BT}(\theta) = \|r(\theta)\|_2^2$$

**The Generalized Gauss-Newton Approximation**  The exact Hessian of such a loss is $\nabla^2 L = 2J_r^\top J_r + 2\sum_i r_i \nabla^2 r_i$. The Gauss-Newton method discards the second-order term (which involves the Hessian of the residuals), assuming the residuals are small or the function is locally linear. The approximate Hessian is

$$\mathbf{H}_{GN}^{BT} = 2\mathbf{J}_r^\top \mathbf{J}_r$$

where $\mathbf{J}_r = \frac{\partial r}{\partial \theta} \in \mathbb{R}^{h^2 \times mh}$ is the Jacobian of the residual vector with respect to the parameters.

However, explicitly forming $\mathbf{J}_r^\top \mathbf{J}_r$ is computationally prohibitive ($O((mh)^2)$). We compute the Hessian-Vector Product (HVP) for an arbitrary direction vector $v \in \mathbb{R}^{m \times h}$ using automatic differentiation

$$\mathbf{H}_{GN}^{BT} v = 2\mathbf{J}_r^\top (\mathbf{J}_r v)$$

We implement this as a two-step procedure

1. JVP (Forward Mode): Computing the directional derivative of the residuals

$$u = \mathbf{J}_r v = \text{jvp}(r, \theta, v)$$

2. VJP (Reverse Mode): Propagating the result back through the transpose Jacobian

$$\mathbf{H}_{GN}^{BT} v = 2 \cdot \text{vjp}(r, \theta, u)$$

## C.2   SIMCLR (SC)

For SimCLR, the loss is not a sum of squares but a Cross-Entropy (CE) loss. The standard Gauss-Newton approximation for CE is defined by the decomposition of the Hessian into the structure $\mathbf{J}^\top \mathbf{Q} \mathbf{J}$, where $\mathbf{Q}$ is the Hessian of the CE loss with respect to the logits.

Let $\ell(\theta) \in \mathbb{R}^{2n \times 2n}$ be the matrix of pairwise similarities (logits), scaled by temperature $\tau$. The loss is the row-wise cross-entropy

$$\mathcal{L}_{SC} = \sum i = 1^{2n} \text{CE}(\mathbf{p}_i, \mathbf{y}_i) = -\sum i = 1^{2n} \log(\mathbf{p}_i, \mathbf{y}_i)$$

where $\mathbf{p}_i = \text{softmax}(\ell_{i,:})$ is the probability vector for sample $i$.

**The Generalized Gauss-Newton Approximation**   Let $\phi$ map parameters to logits $l = \phi(\theta)$, and let $L$ be the loss composed with $\phi$. The exact Hessian is

$$J_\phi^\top \, \nabla_\ell^2 L \, J_\phi \; + \; \sum_h (\nabla_\ell L)_h \, \nabla_\theta^2 \phi_h.$$

The GGN approximation drops the second term, retaining the curvature of the loss function

$$\mathbf{H}_{GN}^{SC} = \mathbf{J}\phi^\top \mathbf{Q} \mathbf{J}_\phi$$

Where $Q = \nabla_l^2 L$ is a block-diagonal matrix and each block corresponds to the Hessian of the softmax-cross-entropy for a single row $i$ with probability vector $\mathbf{p}_i$. The corresponding Hessian block is

$$\mathbf{Q}_i = \mathrm{diag}(\mathbf{p}_i) - \mathbf{p}_i \mathbf{p}_i^\top$$

To apply the preconditioner $\mathbf{H}_{GN}^{SC} v = \mathbf{J}_\phi^\top \mathbf{Q} \mathbf{J}_\phi v$, we utilize the chain rule structure inherent in the GGN

1. JVP (Forward Mode): Computing the perturbation in logits given perturbation in parameters $v$
$$\delta\ell = \mathbf{J}_\phi v = \mathrm{jvp}(\phi, \theta, v)$$

2. **Q** (Softmax Geometry): Applying the Hessian of the CE loss to the logit perturbation. This operation is efficient because **Q** is never materialized; we compute the matrix-vector product row-wise
$$\delta g_i = \mathbf{Q}_i(\delta\ell_i) = \mathbf{p}_i \odot (\delta\ell_i) - \mathbf{p}_i(\mathbf{p}_i^\top \delta\ell_i)$$
where $\delta g$ represents the resulting gradient perturbation.

3. VJP (Reverse Mode): Backpropagating the gradient perturbation to parameter space
$$\mathbf{H}_{GN}^{SC} v = \mathbf{J}_\phi^\top(\delta g) = \mathrm{vjp}(\phi, \theta, \delta g)$$

## D   GENERAL LOSS PRECONDITIONER

We provide the following intuition on the proposed general loss preconditioner; KREPES projects data into a kernel-induced feature space defined by a set of $m$ Nyström landmarks. The Gram matrix $K_{mm}$, computed once per training run, captures the geometry of this landmark space and forms the basis of the learned representation. Since the curvature of the loss with respect to the projection matrix $\tilde{A} \in \mathbb{R}^{m \times h}$ is fundamentally linked to this geometric structure, $K_{mm}$ serves as a natural and computationally efficient candidate for preconditioning. Preconditioning with $K_{mm}$ effectively performs gradient updates in the geometry induced by the kernel, aligning the optimization dynamics with the underlying structure of the representation space.

Algorithm 4 elaborates the procedure needed to update the parameters $\tilde{A}$ with the general loss preconditioner discussed in 3.2.

---

**Algorithm 4** General Loss Preconditioner

---

1: **procedure** GENERALPC($\nabla\tilde{A}, K_{mm}, \lambda, T$)
2:    $\triangleright \nabla\tilde{A} \in \mathbb{R}^{m \times h}$ is the gradient matrix
3:    $g_p \in \mathbb{R}^{m \times h}$
4:    $\mathrm{mvp}(v) = (K_{mm} + \lambda I)v$                    $\triangleright$ Define matrix-vector product function
5:    **for** $j \leftarrow 1$ to $h$ **do**                    $\triangleright$ Apply CG to each column of the gradient
6:        $g_j \leftarrow \mathrm{column}(\nabla\tilde{A}, j)$
7:        $(g_p)_j \leftarrow$ CONJUGATEGRADIENT($\mathrm{mvp}, g_j, T, 10^{-6}$)
8:    **end for**
9:    $\nabla\tilde{A} \leftarrow g_p$                    $\triangleright$ Update gradient in-place
10: **end procedure**

---

# E   NYSTRÖM INITIALIZATION

Algorithm 5 shows the process of initializing the parameters discussed in 3.1.

---

**Algorithm 5** Principal Component Initialization (PCI)

---

1: **procedure** PCI($K_{mm} \in \mathbb{R}^{m \times m}, h, \tilde{A}_0 \in \mathbb{R}^{m \times h}$)
2:    $\triangleright$ Considering Nyström approximation $K_{nn} \approx K_{nm} K_{mm}^\dagger K_{mn}$
3:    Rank-$h$ eigendecomposition of $K_{mm}$, $U_h \in \mathbb{R}^{m \times h}$, $\Lambda_h \in \mathbb{R}^{h \times h}$ s.t. $K_{mm}^\dagger \approx U_h \Lambda_h^{-1} U_h^\top$
4:    $\triangleright$ Define $\Phi = K_{nm} U_h \Lambda_h^{-1/2} \in \mathbb{R}^{n \times h}$, s.t. $K_{nn} \approx (K_{nm} U_h \Lambda_h^{-1/2})(K_{nm} U_h \Lambda_h^{-1/2})^\top = \Phi\Phi^\top$
5:    Compute the initial parameter matrix
6:    $\tilde{A}_0 \leftarrow U_h(\Lambda_h + \epsilon \mathbf{I}_h^{-1/2})$
7:    **return** $\tilde{A}_0$
8: **end procedure**

---

## E.1   ABLATION STUDY ON PARAMETER INITIALIZATION

In table 4 we report the ablation study results regarding our proposed Principal Component Initialization method and the random Kaiming initialization from He et al. (2015).

Table 4: Downstream classification test accuracy across three datasets and three losses with random Kaiming initialization and our proposed Principal Component Initialization (PCI) method.

| Dataset | BT | | SimCLR | | BYOL | |
|---|---|---|---|---|---|---|
| | Kaiming | PCI | Kaiming | PCI | Kaiming | PCI |
| Adult-1M | 83.49 | 83.78 | 83.68 | 83.67 | 83.34 | 83.76 |
| CoverType | 68.98 | 71.32 | 53.00 | 75.17 | 67.94 | 70.31 |
| CIFAR-10 | 89.00 | 90.31 | 89.22 | 89.54 | 87.81 | 88.76 |

# F   ABLATION STUDY ON DIFFERENT LANDMARK SELECTION METHODS

In table 5 we report the downstream performance on the MNIST dataset, across all the proposed loss functions in the paper with three different landmark selection algorithms.

Table 5: Downstream classification test accuracy across all the proposed losses on MNIST dataset with three different algorithms for landmark selection.

| | Random Uniform | Kmeans++ | Leverage Score |
|---|---|---|---|
| BT | 94.95 | 97.41 | 97.95 |
| VICReg | 90.18 | 89.57 | 89.69 |
| BYOL | 79.53 | 84.68 | 85.30 |
| SimCLR | 95.35 | 98.90 | 95.75 |
| Spectral Cont. | 94.22 | 97.99 | 98.09 |
| Simple Cont. | 88.67 | 89.64 | 89.34 |
| KAE | 96.42 | 98.89 | 98.60 |
| KPCA | 89.12 | 97.62 | 97.63 |

# G   PERFORMANCE AND TIME COMPARISON: FULL KERNEL VS KREPES

Figure 4 compares the downstream classification accuracy and training time of the full kernel against KREPES using $m = O(\sqrt{n})$ landmarks on varying subset sizes of the Adult dataset with an RBF

Table 6: CIFAR-10 (KREPES is without preconditioner).

| Loss | Acc (KREPES) | Acc (NN) | Train Time (KREPES) | Train Time (NN) |
|---|---|---|---|---|
| BT | 89.11 | 85.32 | 0.0578 | 1.119 |
| SimCLR | 89.17 | 84.82 | 0.0028 | 0.321 |
| VICReg | 88.69 | 84.52 | 0.2063 | 0.671 |
| BYOL | 88.76 | 78.12 | 0.0980 | 0.428 |

Table 7: Inference times for CIFAR-10.

| Loss | Inference (KREPES) | Inference (NN) |
|---|---|---|
| BT | 0.00016 | 0.4404 |
| SimCLR | 0.00014 | 0.0444 |
| VICReg | 0.00012 | 0.0442 |
| BYOL | 0.00014 | 0.0442 |

kernel and the Barlow-twins pretraining loss. The results show that KREPES achieves accuracy comparable to the full kernel while requiring only a fraction of the computational time. This highlights that our approximation preserves the predictive power of the full kernel, despite operating at substantially reduced complexity, thereby enabling kernel-based representation learning to scale efficiently to larger datasets.

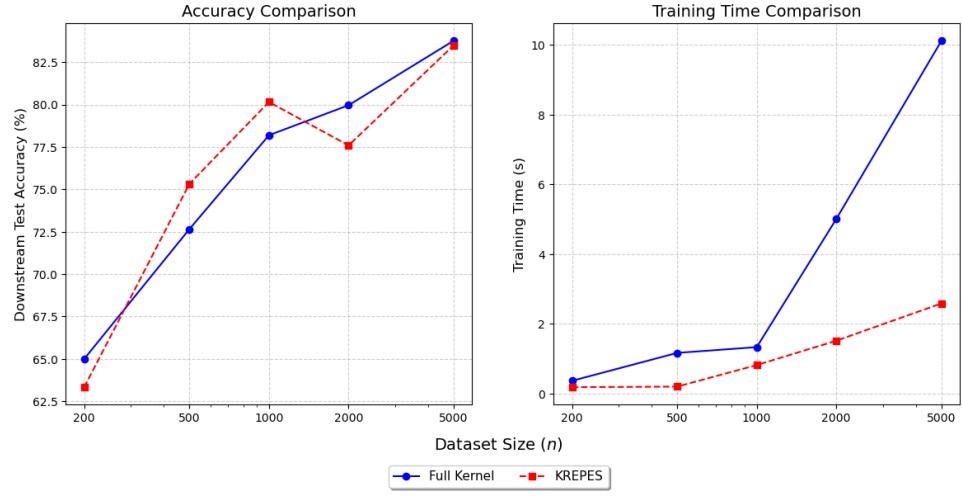

Figure 4: Downstream classification accuracy vs. training time comparison between KREPES with $m = O(\sqrt{n})$ landmarks, and the full kernel for different subset sizes of the Adult dataset (with the RBF kernel, and Barlow-twins pretraining loss). As depicted KREPES is much faster in time and the performance drop is only negligible.

## H  ABLATION STUDY: EFFECT OF PRECONDITIONERS

In tables 6, 7, 8, 9 performance and training time of KREPES without a preconditioner together with the corresponding neural network is depicted. We report results for the identical setup without our preconditioner. We observe that removing the preconditioner leads to a minor drop in accuracy, but reduces training time substantially, making KREPES faster than the neural network baseline while still maintaining competitive or superior accuracy. We are additionally providing the inference time for CIFAR-10 which shows significantly faster values for KREPES than the neural network.

Table 8: Adult (KREPES is without preconditioner).

| Loss | Acc (KREPES) | Acc (NN) | Train Time (KREPES) | Train Time (NN) |
|---|---|---|---|---|
| BT | 83.58 | 83.72 | 0.044 | 0.021 |
| SimCLR | 81.67 | 83.79 | 0.148 | 0.036 |
| VICReg | 83.40 | 82.65 | 0.105 | 0.022 |
| BYOL | 83.36 | 83.50 | 0.010 | 0.024 |

Table 9: Covertype (KREPES is without preconditioner).

| Loss | Acc (KREPES) | Acc (NN) | Train Time (KREPES) | Train Time (NN) |
|---|---|---|---|---|
| BT | 68.83 | 73.71 | 0.231 | 0.021 |
| SimCLR | 52.67 | 68.10 | 0.015 | 0.019 |
| VICReg | 72.85 | 71.08 | 0.169 | 2.640 |
| BYOL | 70.78 | 72.36 | 0.004 | 0.240 |

## I CONCEPTUAL INFLUENCE VIA CONCEPT ACTIVATION VECTORS

In order to obtain the concept activation vector $v_c$, we follow the procedure in Kim et al. (2018). First we need to gather the concept positive and negative datasets, one with the concept present in them and the other without the concept. For our example in this paper we choose the concept *"Sea"* from the class Sea in the CIFAR-100 dataset. Hence, all the samples in this class which are 500 images, will be the positive examples for the concept Sea. Next we have the negative examples from the domain on which our representation learning model KREPES was trained on. We pick 200 random images from a diverse set of classes: cat, dog, horse, truck, automobile.

We compute representations for all the examples, using our trained KREPES we get the $k$-dimensional representation for every image we gather in as positive and negative examples for concept Sea. We define the representation of the positive examples for the concept as

$$Z_{concept+} = K_{cm}\tilde{A}$$

where $K_{cm}$ is the eNTK between the concept positive examples and the landmarks. Same way we calculate the representation of the random negative examples. Further, we train a linear binary classifier which has to learn the direction that separates "Sea" from "random". For this purpose we use a linear SVC classifier from sklearn library. The concept vector $v_c$ is then simply the normalized weight vector learned by the classifier. This vector points from the "random" examples towards the "Sea" examples.

## J COMPUTATIONAL RESOURCES

The experiments are conducted on a high-performance computing (HPC) cluster equipped with NVIDIA H100 and A100 GPUs. For training, we primarily use a single NVIDIA H100 GPU. Depending on the model size, we employed up to four GPUs to compute the eNTKs in parallel. For example, computing the eNTK for ResNet34 on CIFAR-10 using four GPUs takes less than 2 hours.

**LLM Usage** In this paper, LLMs are used to refine the writing, improve the clarity of plots, and provide suggestions on hyperparameter ranges. They also assisted with debugging the code and in searching for some of the related works and references, although all suggestions were carefully verified by the authors.

