# OpenReview forum: "Interpretable Kernel Representation Learning at Scale: A Unified Framework Utilizing Nyström Approximation"
_ICLR.cc/2026/Conference — Submitted to ICLR 2026_

### Official Review · Reviewer_9bGE · 2025-10-24

**Soundness:** 3
**Presentation:** 2
**Contribution:** 2
**Rating:** 4
**Confidence:** 5

**Summary:**

This paper proposes a scalable framework, called KREPES, for kernel-based representation learning.  This framework relies on the Nystrom approximation, such that a representation f(x) lies in the span of some anchor points functions, say phi(tilde{x}_i), where the points \tilde{x}_i are called landmarks. Unlike the classical Nyström method, KREPES does not minimizes the residual in the RKHS norm. Instead, it allows for alternative self-supervised loss functions, which are optimized using gradient-based methods. The paper further explores preconditioning strategies and scalable techniques for efficient landmark selection.

**Strengths:**

- The overall pipeline makes sense and the combination of kernel representation with self-supervised learning is novel and interesting. That being said, it may be useful to position the paper with respect to other work relying on kernel and self-supervised learning such as  Zheng et al. Self-supervised learning with rotation-invariant kernels. ICLR 2023.
 - The code is provided and seems efficient.

**Weaknesses:**

- The empirical validation is not convincing. Using self-supervised loss functions within the Nystrom approximation is new, but we are missing an experimental case where the formulation/method is actually useful. Scalability is demonstrated in Section 4, to some extent, but the reported numbers are actually very far from the state of the art. I am also confused by the baseline NN, which performs particularly badly (60% accuracy on CIFAR-10 is **very** low). Why is that? How is NN trained? on which data? with which loss function? It is of course not directly comparable, but in the context of self-supervised learning, we are used to see extremely high accuracies on CIFAR-10 (e.g., 99% accuracy for DINO when pretrained on ImageNet1k).

  - It may be useful to provide computational time that include kernel precomputation (and report breakdown: kernel computation vs optimization). I was a bit lost by what average time means in Table 1.

  - The claims about interpretability are to be taken with caution. The decomposition is indeed explicit, but it does not necessarily translate
    into human-understandable features. It would strengthen the paper to show case studies where one inspects landmark contributions and relates them to meaningful patterns beyond the experiment on CIFAR-10 which is rather toyish.

**Questions:**

- is there any hope to extend existing theoretical results about Nystrom approximation to the loss functions used in this paper?

  - by changing the loss in the Nystrom method, the resulting optimization problems become nonconvex. Other difficulties may arise such as loss function with trivial solutions (called collapsed solutions in the SSL literature). Is it a problem within the KREPES framework?

---

> ### Author Response · Authors · 2025-11-23
> **General Context and Response to W1**
>
> We thank the reviewer for their comments. We respond to them below and clarify a few points. However, first, we provide the context of this paper, which explains the significance of the present work. It is important to note that the paper does not aim to develop SSL models for vision/language tasks. It is well known that deep models are essential in such domains. In contrast, it is not evident why large tabular models (that is, SSL models for tabular data) are primarily deep architectures. Hence, the present paper explores alternatives in terms of kernel SSL models, which provide the added advantage of interpretability (in fact, Section 5 is a major contribution of our work, which the reviewer overlooked). Note that the standard benchmarks in the tabular SSL literature contain up to a few million samples, which is smaller than large-scale image/language benchmarks, but still quite large for naive kernel implementations. Hence, it is important to scale up kernel SSL to millions of samples, which is the core technical contribution of this work. As we show, efficient kernel SSL implementations are as good as deep SSL models in all tabular datasets (as well as CIFAR-10), while additionally providing interpretability.
>
> ## W1
> We believe the reviewer’s expectation of 90%+ accuracy may stem from results reported in prior SSL works such as Chen et al. (2020), Bardes et al. (2022), and Grill et al. (2020), all of which use pretrained ResNet-50 architectures for CIFAR-10. Our setting differs in several crucial ways, all of which are documented in Appendix B but may have been overlooked:
>
>    1. Linear classifier uses only 10% of labeled data; as explained in Appendix B, our evaluation protocol does not use the full labeled CIFAR-10 training set. After self-supervised training, the linear classifier is trained on only 10% of the labeled data. This differs significantly from standard linear probing protocols, which typically use all labels.
>    2. No supervised fine-tuning; we strictly evaluate representations via linear classification without any supervised fine-tuning, whereas many SSL papers fine-tune or implicitly improve representations through heavy augmentations.
>    3. Much lighter augmentation pipeline; our augmentation pipeline is intentionally simpler and less diverse (also depicted in details in appendix B) than those used in the above-mentioned works, which is known to affect absolute accuracy.
>    4. No projection head for the neural network baseline; to ensure a fully fair comparison with KREPES, we also do not use any projection head in the neural network baseline. In contrast, all high-performing SSL works (including SimCLR, BYOL, Barlow Twins, and DINO) rely critically on multi-layer projection heads, which improve performance.
>    5. Equal conditions for KREPES and neural network baselines; most importantly, all of these constraints apply equally to KREPES and the neural network baselines. Thus, while the absolute accuracy is lower than typical ResNet-50 SSL results reported in the literature, the comparison between the two models under our protocol is fair and controlled.
>    6. Updated CIFAR-10 results; during revision, we identified an implementation error in the CIFAR-10 baseline code. We **updated Table 1** accordingly in the revised manuscript.
> This discussion is added to **appendix B.4**. We report ablation results in **appendix H** for the identical setup without preconditioner. We observe that removing the preconditioner leads to a minor drop in accuracy, but reduces training time substantially, making KREPES faster than the neural network baseline while still maintaining competitive or superior accuracy. We are additionally providing the inference time for CIFAR-10 which shows significantly faster values for KREPES than the neural network.
>
> However, even in the mentioned approaches in the cases they have not done any supervised fine-tuning, they don't report higher that ~ 91% on CIFAR-10 with pretrained ResNet50. Furthermore, we draw the reviewer's attention to the fact that it is not a fair comparison to put our shallow linear kernel based framework next to extremly complex deep frameworks such as DINO.
>     A more appropriate comparison is with prior kernel-based representation learning works. When contrasted with supervised kernel methods such as Arora et al. (2019), Novak et al. (2019), Mairal et al. (2016), and Shankar et al. (2020), all of which are not scalable and not self-supervised, our method achieves performance that is competitive with the state of the art under comparable architectural restrictions. Importantly, KREPES reaches this level in a purely self-supervised manner while scaling to datasets far larger than those typically handled by kernel SSL methods.
>
> Regarding training details, the full architecture and training protocol were documented in Appendix B. Please refer to our **Global Response** which points to more added details in our revised version.

---

> ### Author Response · Authors · 2025-11-30
> **Response to W2, Q1, Q2**
>
> ## W2
> The reviewer makes a philosophical assumption that only human-understandable features of data (such as ones used in decision trees) is the only way to develop interpretable models. Similarities with pre-selected samples or ranking with respect to similarities also provide interpretability. There are earlier works arguing kernel SVM is more interpretable than neural networks (arXiv:1704.05848), while recent deep learning literature suggests approximating trained neural networks with kernel machines (arXiv:2305.20002, arXiv:1811.09720, arXiv:1802.09210). The philosophical message of this work is that if kernel SSL methods can be scaled up and provide reasonable performance, we should simply train  shallow (kernel) models instead of approximating trained NNs with kernels.
>
> ## Q1
> The KREPES framework allows one to incorporate quite general SSL loss functions, and we do not expect that general statistical guarantees for Nyström approximation can be derived for loss functions like SimCLR. However, we certainly believe that guarantees could be derived for some specific SSL losses.  For instance, arXiv:2309.02028 shows that the solution of kernel SSL under  a simple contrastive loss can be analytically derived as a spectral embedding, and one could potentially extend some known results for Nyström kernel PCA to this case.
>
> On the other hand, although the Barlow Twins is quite simple, it results in a polynomial of degree 4, which is neither convex nor Lipschitz (which is currently the most general setting where Nyström approximation guarantees are known (arXiv:2006.10016).  Yet, arXiv:2411.11176 analyzes the Barlow twins loss in the NTK regimes and provide some ideas on how to tackle such loss functions. We believe that guarantees  can be derived for this loss as well, but it is beyond the focus of the current paper.
>
> Nonetheless, if we aim to sustain the relevance of kernel methods in the age of deep learning / foundation models, we need to focus on developing general-purpose large scale kernel software, which is the focus of this paper. We would like to point to the Eigenpro 3.0 paper (Abedsoltan et al, ICLR 2023), which also focuses on large scale kernel regression under the squared loss, but does not provide any statistical guarantee.
>
> ## Q2
> Trivial (collapsed) solution naturally arises in the SSL formulation, and not specific to any model (neural network or kernel). Indeed collapse can occur if one uses a naïve SSL loss in the KREPES framework. However, most of the popular SSL losses incorporate mechanisms to avoid collapse, such as normalization or covariance regularization in SimCLR/VICReg or use of stop-grad or exponential moving average in BYOL. In the KREPES framework, we already incorporate these terms in the loss. Hence, KREPES does not suffer from collapsed solutions any more than a deep learning architecture would.

---

### Official Review · Reviewer_YmzL · 2025-10-29

**Soundness:** 1
**Presentation:** 2
**Contribution:** 2
**Rating:** 4
**Confidence:** 3

**Summary:**

This paper presents KREPES, a scalable framework for kernel-based self-supervised representation learning. The key idea is to apply Nyström approximation to reduce the computational cost of kernel methods so they can operate on large datasets. The framework supports multiple SSL objectives (e.g., SimCLR, BYOL, VICReg, Barlow Twins) and is optimized via gradient-based training on a reduced set of landmark points. The authors propose a PCA-based initialization and second-order preconditioning strategies to improve optimization efficiency. Experiments on some tabular and vision datasets are presented. Additionally, the model naturally supports interpretability via representer landmarks, enabling analysis of influential samples and concept-level explanations.

**Strengths:**

-The paper tackles an important and timely challenge: enabling scalable kernel-based representation learning in the self-supervised setting. This represents a meaningful contribution.

-The proposed framework is flexible and unifies a wide range of SSL objectives.

-The interpretability component is interesting. Leveraging representer points to analyze landmark influence and concept alignment provides clear explanatory benefits compared to standard deep SSL models.

-The empirical results suggest that the approach can be competitive across different data modalities, particularly in settings where interpretability is a priority.

**Weaknesses:**

-The literature review on Nyström methods is quite outdated, and many recent works are not mentioned.

-Lines 190–193 discuss Rudi et al. (2017), which itself builds on Less Is More (Rudi et al., 2015). However, this earlier work shows that
m=O(\sqrt{n}) landmarks are required under standard assumptions (not O(logn) as the paper implies).

-In lines 239–240, the authors correctly point out that their setting differs from Rudi et al. (2017) because no closed-form solution exists for these SSL losses. However, the literature review remains incomplete, as other works addressing Nyström approximation for general losses—such as for example, "The Nyström method for convex loss functions" by Della Vecchia et al. (2024) or “Fast and Accurate Refined Nyström Based Kernel SVM” by Li et al. (2015) and others—are not cited.

-Again, in lines 318–320, while discussing landmark selection, only Rudi et al. (2017) is referenced. Recent work aiming to mitigate this bottleneck, such as "On fast leverage score sampling and optimal learning" by Rudi et al. (2018) with the BLESS algorithm, is missing.

-Some choices are quite obscure and not well-motivated, for example at lines 276-279 where the preconditioner used in FALKON is mentioned but a simpler one is indeed used because "empirically it improves optimization ... ". Does this hold in general? Is it limited to the analyzed datasets and losses?

-The experiments in Section 4 are somewhat limited, and the results are not fully convincing. For instance, it is stated that the neural network baselines are “generally faster”, but no discussion is given on the magnitude of this gap — e.g., on CoverType the NN appears to be up to 1000× faster, which is substantial. Similarly, on CIFAR-10 the proposed method outperforms the NN baselines by up to 40 accuracy points, which seems extreme and suggests a potential issue with the fairness or quality of the NN baselines.

**Questions:**

Some perplexities are expressed above. Additionally:

Table 1 shows differences of up to 40 accuracy points between the proposed method and the neural network baselines. How is such a large performance gap possible? Were the neural networks trained correctly with standard and competitive SSL protocols? Is the comparison fair?

-Prior work (e.g., Rudi et al., 2015; Della Vecchia et al., 2024, among others) provides both theoretical and empirical guidance on properly selecting the number of Nyström landmarks m, typically
m=O(\sqrt{n}), to ensure near-optimal accuracy while reducing computation. In the present paper, however, no theoretical guarantees are provided for the chosen m; the experimental results suggest the opposite of the expected trade-off: the Nyström-based model seems to achieve higher accuracy (sometimes significantly so), while being 100× to 1000× slower than the suggested baseline; the procedure for selecting m is not discussed. A deeper clarification and justification of these aspects is needed to support the claims regarding scalability and fairness of the evaluation.

I will reconsider my rating once the authors have addressed the questions above.

---

> ### Author Response · Authors · 2025-11-23
> **Response to W1, W2, W3, W4, and Q1**
>
> We thank the reviewer for the comments. Regarding the reviewer's concern of our presentation we have included
> - **Algorithm 1**: KREPES: Kernel Representation Learning at Scalable via Nyström Approximation, which more clearly portraits all the steps of our proposed framework and highlights the novelties of our contribution.
> - **Algorithm 2**: KREPES Interpretability: Influence Scores and Conceptual Profiles, which provides the explicit procedure of our interpretability analysis, with the hope of drawing more attention to section 5.
> - Appendix C **full derivation of GGN preconditioners**
>
> ## W1-4:
> 1. The expression $O(\log n)$ in our manuscript refers to the number of iterations $t$ and not the scaling for $m$, which should be $m = O(\sqrt{n})$. We corrected the sentence in the revised manuscript.
>
> 2. , 3. We thank the reviewer for the references; all are added in section 3 (1,2 in paragraph "From Analytic to Algorithmic Design" and last one in section 3.3).
>
> 4. We provide the following intuition which we incorporated in the revision together with an algorithm of the general loss preconditioner in **appendix D**. KREPES projects data into a kernel-induced feature space defined by a set of $m$ Nyström anchor points. The Gram matrix $K_{mm}$, computed once per training run, captures the geometry of this landmark space and forms the basis of the learned representation. Since the curvature of the loss with respect to the projection matrix $A \in \mathbb{R}^{m \times h}$ is fundamentally linked to this geometric structure, $K_{mm}$ serves as a natural and computationally efficient candidate for preconditioning. Preconditioning with $K_{mm}$ performs gradient updates in the geometry induced by the kernel, aligning the optimization dynamics with the underlying structure of the representation space. This geometric justification, together with the empirical evidence already provided, supports the use of our general-loss preconditioner beyond the MSE setting.
>
>     Our empirical findings support these choices. As shown in the paper, the loss-specific GGN-based preconditioners generally outperform the general preconditioner for the objectives we study (BT and SimCLR). At the same time, we observe that the general-loss preconditioner improves optimization performance relative to no preconditioning at all, in most of the datasets and loss functions in our experiments (we only observed worse performance when using the general loss preconditioner in the case of MNIST).
>
> ## Q1:
> We note that longer training times arise from the number of iterations required by the CG solver used for preconditioners. We report ablation results in **appendix H** for the identical setup without preconditioner, showing that it leads to a minor drop in accuracy, but reduces training time substantially, making KREPES faster than the NN baseline while still maintaining competitive or superior accuracy. We additionally provided the inference time for CIFAR-10 which shows significantly faster values for KREPES than the neural network.
>
> We believe the reviewer’s expectation of 90%+ accuracy may stem from results reported in works such as Chen et al. (2020), Bardes et al. (2022), and Grill et al. (2020), all of which use pretrained ResNet-50 architectures for CIFAR-10. Our setting differs in several crucial ways, all of which are in Appendix B:
> 1. Linear classifier uses only 10% of labeled data; as explained in Appendix B, our evaluation protocol does not use the full labeled CIFAR-10 training set. After self-supervised training, the linear classifier is trained on only 10% of the labeled data. This differs significantly from standard linear probing protocols, which typically use all labels.
> 2. No supervised fine-tuning; we strictly evaluate representations via linear classification without any supervised fine-tuning, whereas many SSL papers fine-tune or implicitly improve representations through heavy augmentations.
> 3. Much lighter augmentation pipeline; our augmentation pipeline is intentionally simpler and less diverse (also depicted in details in appendix B) than those used in the above-mentioned works, which is known to affect absolute accuracy.
> 4. No projection head for the neural network baseline; to ensure a fully fair comparison with KREPES, we also do not use any projection head in the NN baseline. In contrast, all high-performing SSL works rely critically on multi-layer projection heads.
> 5. Equal conditions for KREPES and neural network baselines; most importantly, all of these constraints apply equally to KREPES and the neural network baselines. Thus, while the absolute accuracy is lower than typical ResNet-50 SSL results reported in the literature, the comparison between the two models under our protocol is controlled.
> 6. Updated CIFAR-10 results; during revision, we identified an implementation error in the CIFAR-10 baseline code. We **updated Table 1** accordingly in the revised manuscript.
> This discussion is added to **appendix B.4**.

---

> ### Author Response · Authors · 2025-11-30
> **Response to Q2**
>
> ## Q2:
> We appreciate the reviewer’s comments. First, regarding the observation that KREPES sometimes achieves higher accuracy than the baselines: this arises because our comparisons differ fundamentally from those in prior scalable-kernel works such as Rudi et al. (2015) or Della Vecchia et al. (2024). These earlier frameworks typically evaluate Nyström approximations against the full kernel solution, and the expected behavior is that using fewer landmarks reduces computational cost at the expense of some accuracy relative to the exact kernel method. We have included additional results showing this type of comparison; plots in **appendix G** of the revised manuscript. We compare the downstream performance and the training time of full kernel with KREPES for different subsets of the Adult dataset. As depicted performance is closely comparable and KREPES is significantly faster.
>
> In contrast, our baselines are not full-kernel methods, but rather the corresponding neural networks from which the empirical NTKs are computed. To the best of our knowledge, there is no existing prior work on fully self-supervised kernel-based representation learning at this scale, so direct comparisons with other kernel SSL frameworks are not available. Consequently, our method is often slower than the neural network baseline and may outperform it; this divergence from prior kernel literature stems from the fundamentally different comparison point, not from a reversal of the conventional Nyström trade-off. We will clarify this distinction more explicitly in the revised manuscript.
>
> Regarding the choice of m: the complete procedure is presented in Section 3.3 under Efficient Landmark Selection. In the revision, we will add explicit algorithm boxes summarizing the procedures in Sections 3.1, 3.2, and 3.3 to make the selection mechanism clearer and more accessible. While we do not provide new theoretical guarantees for selecting m, our approach is guided by standard Nyström principles and is designed to balance representation quality with computational feasibility in the self-supervised setting. We will make this motivation clearer and explicitly connect our selection strategy to the theoretical context provided in prior work.
>
> Regarding the reviewer’s remark that “no theoretical guarantees are provided for the chosen m,” we would like to clarify that offering theoretical guarantees for m-selection in our setting is not feasible within the scope of this work. Importantly, significant related contributions in scalable kernel methods, such as Abedsoltan et al. (2023) also do not provide theoretical guarantees, yet are valuable precisely because they develop the practical Eigenpro 3.0 framework (for the case of least squared regression). In our setting, the challenge is even more structural. Thanks to the generality of KREPES, we handle a variety of SSL loss functions, many of which lack closed-form analytic properties (e.g., non-convexity, non-smoothness, and high-order dependencies). For instance, Barlow Twins induces a fourth-order objective with respect to the parameters, and therefore is neither convex nor Lipschitz, meaning that existing Nyström generalization tools such as those from Della Vecchia et al. (2024), cannot be applied to the SSL losses studied in this paper.
>
> We believe that some theoretical guarantees could be developed under specific SSL losses. For instance, arXiv:2309.02028 shows that the solution of kernel SSL under  a simple contrastive loss can be analytically derived as a spectral embedding, while arXiv:2411.11176 analyzes the Barlow twins loss in the NTK regimes. Ideas from these works can potentially be used to derive the optimal number of Nyström landmarks. A general theory for all SSL losses may not be possible. Nonetheless, if we aim to sustain the relevance of kernel methods in the age of deep learning / foundation models, we need to focus on developing general-purpose large scale kernel software, which is the focus of this paper.

---

### Official Review · Reviewer_Tr1g · 2025-11-04

**Soundness:** 2
**Presentation:** 1
**Contribution:** 2
**Rating:** 2
**Confidence:** 4

**Summary:**

The paper introduces KREPES, a scalable kernel-based self-supervised learning algorithm that optimizes modern SSL losses (SimCLR, BYOL, VICReg, Barlow Twins) directly in kernel space using stochastic gradient descent. It achieves scalability via a Nyström approximation and stabilizes training with a preconditioner inspired by Rudi et al. (2017), extended to non-quadratic losses through Gauss–Newton updates. Using kernels like the empirical NTK of pretrained networks, KREPES matches neural network performance while remaining interpretable and computationally efficient.

**Strengths:**

1. Investigating self-supervised losses within kernel methods is a relatively underexplored area with strong potential.

2. The paper explores a practical approach to scaling kernel models to larger datasets rather than a theoretical advance in scalability.

3. Evaluating multiple modern SSL losses (SimCLR, BYOL, VICReg, Barlow Twins) within one unified kernel framework is important for robustness and broad applicability.

**Weaknesses:**

1. The authors emphasize scalability, but I remain unconvinced. The method builds on a preconditioner from Rudi et al. (2017), which still requires an $O(m^2)$ memory footprint. This scaling becomes impractical when the number of landmarks m grows large. For true scalability, I would expect to see experiments demonstrating how the method performs when m reaches millions of landmarks not just $n >> 10^6$.


2.  If the paper’s goal is to show that allowing diverse self-supervised losses (beyond MSE) leads to better performance, the reported results do not convincingly support that claim. The method uses an **empirical NTK (eNTK)** computed from a pretrained network, but it is unclear which model produced this eNTK. If it was derived from a strong architecture like **ResNet-50**, then even a simple MSE-based kernel regression on those features should yield strong results or even simpler get the last layer embeding of the trained model and use a laplace kernel.


3. Table 1 raises questions — the CIFAR-10 neural network baseline reportedly achieves only about 60–70% accuracy, which seems unusually low for a ResNet-50-sized model (~25 M parameters), as such a network typically exceeds 90% accuracy. Clarifying which neural network was used for both the baseline and the eNTK would greatly help interpret these results.


4.  The main contribution of the paper is not fully clear. It would help if the authors included an explicit algorithm box summarizing what **KREPES** actually does and what is novel about it. As I understand it, KREPES applies stochastic optimization (e.g., Adam) to kernel models using the **preconditioner from Rudi et al.**, while extending it to non-quadratic self-supervised losses. However, that preconditioner was originally derived for the MSE loss in kernel ridge regression; applying it empirically to other, non-convex SSL losses seems somewhat ad-hoc. It would strengthen the paper if the authors could justify why this preconditioner is expected to help under those more general losses, or provide theoretical or empirical support for this choice.

**Questions:**

See weakness.

---

> ### Author Response · Authors · 2025-11-24
> **General Context and Response to W1, W2**
>
> We thank the reviewer for their comments. We respond to them below and clarify a few points that the reviewer misunderstood. However, first, we provide the context of this paper, which explains the significance of the present work. It is important to note that the paper does not aim to develop SSL models for vision/language tasks. It is well known that deep models are essential in such domains. In contrast, it is not evident why large tabular models (that is, SSL models for tabular data) are primarily deep architectures. Hence, the present paper explores alternatives in terms of kernel SSL models, which provide the added advantage of interpretability (in fact, Section 5 is a major contribution of our work, which the reviewer overlooked). Note that the standard benchmarks in the tabular SSL literature contain up to a few million samples, which is smaller than large-scale image/language benchmarks, but still quite large for naive kernel implementations. Hence, it is important to scale up kernel SSL to millions of samples, which is the core technical contribution of this work. As we show, efficient kernel SSL implementations are as good as deep SSL models in all tabular datasets (as well as CIFAR-10), while additionally providing interpretability.
>
> 1. We clarify that our framework includes three preconditioners, only one of which follows the approach of Rudi et al. (2017). The other two, as described detailed in the paper, are based on the Generalized Gauss–Newton method and therefore do not rely on the scalability constraints associated with the Rudi et al. See appendix C for more details of the full derivation of our GGN preconditioners.
>
>     Regarding the concern about scalability with respect to the number of landmarks $m$: the key purpose of KREPES is precisely to reduce the number of required landmarks, thereby enabling kernel methods to scale to large datasets without needing large $m$. Moreover, requesting experiments with m in the order of millions is not meaningful in the Nyström framework. In Nyström, achieving $m \approx 10^6$ landmarks would require working with datasets on the order of $n = \mathcal{O}(m^2)$, i.e., $\approx 10^{12}$ or 100 billion samples! To the best of our knowledge, no publicly available SSL benchmark of this scale exists; some of the largest widely used tabular benchmarks used in tabular foundation model literature, such as the US Census 1990 dataset ($\sim 2.5$M samples), KDD Cup 1999 ($\sim 4$M samples), and Bar Crawl ($\sim 14$M samples)(all from the UCI repository), are several orders of magnitude smaller than the dataset size required to justify $\sim  10^6$ landmarks. Thus, the reviewer’s request is far beyond the scale of any existing benchmark used.
>
>     We therefore believe our experiments, which scale kernel representation learning to datasets of the order of million samples, already represent a previously unattainable extension of kernel scalability, demonstrating that meaningful representation learning with kernels is indeed possible at this scale, which is the central contribution of our work.
>
> 2. We **do not claim using diverse SSL losses leads to better performance**. The reviewer may be aware that MSE is typically not used in SSL, whereas scaling up kernel methods is mostly studied under MSE loss (with some extensions to convex losses). The goal of this paper is to scale up kernels to different SSL losses. By being able to consider various SSL loss functions, we demonstrate that our framework is general and accommodates any user-defined SSL objective. The focus of the paper is on scalability and interpretability of kernel SSL; **not on loss-function comparisons**.
>
>     Regarding the eNTK: As stated in Section 4 and described in Appendix B, all eNTKs (except for CIFAR-10) are computed from **untrained** architectures; eNTK is derived from the same architecture as the neural network baseline, and the appendix elaborates these architectures as well as the exact procedure used to compute the eNTK. We also provide a direct link in the main text pointing to this. For CIFAR-10 specifically, the eNTK is computed using ResNet-34, not ResNet-50. We would appreciate if the reviewer could point us to a scalable performance of over 90\% classificaiton accuracy of an MSE based eNTK model based on a ResNet34 except ours, if they know of any, particularly given that MSE-based training constitutes supervised learning, which is a significantly simpler task than putting a linear classifier on top of the Self-supervised learned representations!
>
>     Finally, it is important to distinguish between two settings. The goal of our work is to provide a shallow, transparent, kernel-based representation-learning framework with only $O(10^5)$ trainable parameters and inherent interpretability (see Section 5) which is fundamentally different from interpreting deep models by applying a Laplace kernel on top of a deep, non-linear, non-transparent network with tens of millions of parameters.

---

> ### Author Response · Authors · 2025-11-30
> **Response to W3, W4**
>
> 3. The architectures used for the neural network baselines and those used to compute the eNTKs are described in detail in Appendix B, and as stated in section 4 of the main paper, for each dataset the eNTK is derived from the same architecture as its corresponding neural network baseline. For CIFAR-10 specifically, the model used is ResNet-34, not ResNet-50.
>
>     We believe the reviewer’s expectation of 90\%+ accuracy may stem from results reported in prior SSL works such as Chen et al. (2020), Bardes et al. (2022), and Grill et al. (2020), all of which use pretrained ResNet-50 architectures for CIFAR-10. Our setting differs in several crucial ways, all of which are documented in Appendix B but may have been overlooked:
>
>     1.	Linear classifier uses only 10\% of labeled data; as explained in Appendix B, our evaluation protocol does not use the full labeled CIFAR-10 training set. After self-supervised training, the linear classifier is trained on only 10\% of the labeled data. This differs significantly from standard linear probing protocols, which typically use all labels.
>
>     2.	No supervised fine-tuning; we strictly evaluate representations via linear classification without any supervised fine-tuning, whereas many SSL papers fine-tune or implicitly improve representations through heavy augmentations.
>
>     3.	Much lighter augmentation pipeline; our augmentation pipeline is intentionally simpler and less diverse (also depicted in details in appendix B) than those used in the above-mentioned works, which is known to affect absolute accuracy.
>
>     4. No projection head for the neural network baseline; to ensure a fully fair comparison with KREPES, we also do not use any projection head in the neural network baseline. In contrast, all high-performing SSL works (including SimCLR, BYOL, Barlow Twins, and DINO) rely critically on multi-layer projection heads, which improve performance.
>
>     5.	Equal conditions for KREPES and neural network baselines; most importantly, all of these constraints apply equally to KREPES and the neural network baselines. Thus, while the absolute accuracy is lower than typical ResNet-50 SSL results reported in the literature, the comparison between the two models under our protocol is fair and controlled.
>
>     6. Updated CIFAR-10 results; during revision, we identified an implementation error in the CIFAR-10 baseline code. We **updated Table 1** accordingly in the revised manuscript.
> This discussion is added to **appendix B.4**.
>     We report ablation results in **appendix H** for the identical setup without preconditioner. We observe that removing the preconditioner leads to a minor drop in accuracy, but reduces training time substantially, making KREPES faster than the neural network baseline while still maintaining competitive or superior accuracy. We are additionally providing the inference time for CIFAR-10 which shows significantly faster values for KREPES than the neural network.
>
> 4. We have added two algorithm boxes to the revised version; **algorithm 1**: KREPES: Kernel Representation Learning at Scalable via Nyström Approximation
> **algorithm 2**: KREPES Interpretability: Influence Scores and Conceptual Profiles
> We also added an algorithm of our parameter initialization in **appendix E**.
>     Regarding the reviewer’s interpretation of the preconditioning strategy, we clarify KREPES incorporates three different preconditioners, only one of which is based on the approach of Rudi et al. (2017). The other two are derived from the Generalized Gauss–Newton (GGN) method, and therefore are not limited to the MSE/KRR setting for which the Rudi et al. preconditioner was developed. These GGN preconditioners are specifically designed to handle non-quadratic, non-convex SSL losses considered in our framework for which we are providing the exact derivation details in the **appendix C**. Although, this part already was included in the main paper (section 3.2), we added more details.
>
>     We had provided empirical support, in Figure 1 (2nd plot), they enable the model to learn higher-rank, and thus more expressive, representations. We provide the following intuition, which we have incorporated in the revision together with an algorithm of the general loss preconditioner in **appendix D**. KREPES projects data into a kernel-induced feature space defined by a set of $m$ Nyström landmarks. The Gram matrix $K_{mm}$, computed once per training run, captures the geometry of this landmark space and forms the basis of the learned representation. Since the curvature of the loss with respect to the projection matrix $A \in \mathbb{R}^{m \times h}$ is fundamentally linked to this geometric structure, $K_{mm}$ serves as a natural and computationally efficient candidate for preconditioning. Preconditioning with $K_{mm}$ effectively performs gradient updates in the geometry induced by the kernel, aligning the optimization dynamics with the underlying structure of the representation space.

---

### Official Review · Reviewer_FoeR · 2025-11-06

**Soundness:** 2
**Presentation:** 3
**Contribution:** 2
**Rating:** 4
**Confidence:** 4

**Summary:**

This paper introduces a scalable framework for kernel-based self-supervised representation learning through Nystrom approximation. The proposed framework achieves comparable learning performance as DNNs with much less training parameters, therefore should be more computationally efficient. The proposed framework also provides some interpretation through representer landmarks. The overall writing is fluent the generalization of kernel methods to a group of loss functions in SSL is appreciated.

**Strengths:**

1. The proposed scalable kernel-based method for representation learning is novel.
2. The improvement of the algorithm through second-order method with pre-conditioner and Hessian approximation is effective.
3. The writing is relatively good.

**Weaknesses:**

1. The paper claims to find a scalable method in the data-hungry demands era, which however, is not reflected through their simulation.
2. The theoretical advantage of kernel methods in representation learning is also not reflected. Without that, based on the current presentation, what we can see is kernel-based SSL achieves comparable performance as, not better than, neural networks. Combined with the first weakness, the work could not persuade us to choose kernel based method over DNNs.
3. The proposed method is claimed to scale efficiently to large dataset, which could be reflected through training time, is missing. Moreover, the test time of the proposed framework in most of the cases (Table 1) is much longer than NN. What would be the benefits choosing KREPES over NN, without explicit theoretical benefits and computation efficiency?

**Questions:**

Please refer to the weakness part.

---

> ### Author Response · Authors · 2025-11-24
> **General Context and Response to W1, W2, W3**
>
> We thank the reviewer for their comments. We respond to them below and clarify a few points that the reviewer misunderstood. However, first, we provide the context of this paper, which explains the significance of the present work. It is important to note that the paper does not aim to develop SSL models for vision/language tasks. It is well known that deep models are essential in such domains. In contrast, it is not evident why large tabular models (that is, SSL models for tabular data) are primarily deep architectures. Hence, the present paper explores alternatives in terms of kernel SSL models, which provide the added advantage of interpretability (in fact, Section 5 is a major contribution of our work, which the reviewer overlooked). Note that the standard benchmarks in the tabular SSL literature contain up to a few million samples, which is smaller than large-scale image/language benchmarks, but still quite large for naive kernel implementations. Hence, it is important to scale up kernel SSL to millions of samples, which is the core technical contribution of this work. As we show, efficient kernel SSL implementations are as good as deep SSL models in all tabular datasets (as well as CIFAR-10), while additionally providing interpretability.
>
> 1. We would like to clarify that, in the context of kernel methods; standard kernels become computationally infeasible even for datasets with a few thousand samples, due to the $O(n^2)$ storage and $O(n^3)$ time complexity associated with kernel matrix computations. Our contribution addresses precisely this classical limitation for SSL losses, as it was addressed for regression in Rudi et al. (2017), Abelsoltan et al. (2023) and related works. With the proposed Nyström-based framework it becomes feasible to train kernel representation-learning models on datasets with order of million samples, which is far beyond the practical regime of conventional kernel methods. Importantly, our scalability analysis show, despite relying on a low-rank approximation, the learned representations retain strong performance. This is the sense in which our method is scalable.
>
> 2. In our work, this advantage is reflected through the interpretability enabled by our kernel-based framework. Building on insights from Yeh et al. (2018), who demonstrated how the representer theorem can be leveraged to interpret neural network predictions, we show that in our setting, where the model is explicitly kernel-based, representer point interpretation is inherent. This provides principled, transparent explanations of the learned representations without requiring additional post-hoc tools. Section 5 of the paper presents detailed interpretability analysis demonstrating this advantage in practice. With respect to the reviewer’s comment that kernel-based SSL “achieves comparable performance” we would like to stress that, this is precisely the contribution we aim to highlight. The goal of the paper is not to outperform deep models, but to investigate whether shallow, transparent models can serve as viable approaches for self-supervised representation learning. Our results show that kernel methods can indeed learn meaningful SSL representations, narrowing the gap with deep neural networks while offering interpretability that deep models do not natively provide. Thus, the contribution lies in demonstrating that deep architectures are not the only option for SSL, and that one can obtain competitive performance while benefiting from the transparency and theoretical grounding inherent to kernel methods (which our work provides great motives for investigating this angle as a future work).
>
> 3. The times reported in Table 1 correspond to training time—**not test time**. This is stated in the table caption (“average time of one forward and backward pass multiplied by the number of epochs required for convergence”). Thus, the training-time analysis is indeed included. The longer training times arise primarily from the number of iterations required by the conjugate gradient solver used within our preconditioning scheme. In appendix H of the revised manuscript we provide examples of a few cases in which we did not use any preconditioner and still achieved comparable results while being significantly faster.
>
> Regarding the benefits of choosing KREPES, these lie in its conceptual simplicity and, crucially, its principled interpretability. As demonstrated in Section 5, our framework enables transparent representer theorem-based explanations that are native to kernel methods and difficult to obtain for deep models without additional post-hoc tools. Combined with the fact that KREPES achieves comparable performance to deep SSL models, our results indicate that kernel-based approaches are a viable for representation learning. This is particularly true for tabular data, where We believe this opens the door to further theoretical developments and provides strong motivation for continued exploration of kernel methods in SSL.

---

### Author Response · Authors · 2025-11-30
**Global Response**

We thank all the reviewers for their feedbacks. We have carefully considered their comments and revised our work to address the raised concerns. We have uploaded a revised version of the paper that includes the following updates:

- **Algorithm 1**: KREPES: Kernel Representation Learning at Scalable via Nyström Approximation, which more clearly portraits all the steps of our propsoed framework and highlights the novel aspects of our contribution.
- **Algorithm 2**: KREPES Interpretability: Influence Scores and Conceptual Profiles, which provides the explicit procedure of our interpretability analysis, with the hope of drawing more attention to section 5 which is a significant part of our contribuiton, and currently seems to have been overlooked by some of the reviewers.
- **Updated CIFAR-10 results**: during revision, we identified an implementation error in the CIFAR-10 baseline code. We updated (last row corresponding to CIFAR-10 of) Table 1 accordingly in the revised manuscript. Further, we provide inference times of both KREPES and NN in appendix H, depicting much faster times for KREPES.
- Appendix B.4 **Discussion on performance difference of SOTA deep models and our neural network baseline on CIFAR-10**
- Appendix C **full derivation of GGN preconditioners**: we are providing the exact derivation details in the appendix C. Although, this part was already included in the main paper (see section 3.2), we added more details.
- Appendic D **Discussion and algorithm 4**: We provided algorithm 4 of our **general loss preconditioner**, as well as the clarifying intuition behind it.
- Appendix E **algorithm 5**: Principal Component Parameter Initialization
- Appendix G Performance and time comparison plots of the **exact kernel** and the **Nyström approximated (via KREPES) kernel**
- Appendix H Tables 6, 7, 8, 9 reporting results of performance and time comparison between **KREPES without preconditioner** and the neural network baseline across datasets, to show **Effect of Preconditioners**; much faster training time and minor drop in performance
- References
    - Della Vecchia et al. The Nyström method for convex loss functions. JMLR, 2024
    - Li et al. Fast and Accurate Refined Nyström Based Kernel SVM. AAAI, 2015
    - Rudi et al. On fast leverage score sampling and optimal learning. Neurips, 2018

     are added in section 3 (first two in the paragraph "From Analytic to Algorithmic Design" and last one in section 3.3).

---

### Meta-Review · Area_Chair_dBbe · 2026-01-06

**Summary:**

The paper proposes a unified, scalable framework for kernel-based representation learning via Nyström approximation called  KREPES, which supports a range of unsupervised and self-supervised losses. Empirical experiments are provided to illustrate the efficiency of the method.

Reviewers generally appreciate that the proposed method is novel and has potential. However, they have many concerns over the current manuscript, including unconvincing claims of scalability and lack of theoretical analysis and guarantees.

**Reviewer Concerns:**

Reviewers have many concerns on the paper, some of which include

- Theoretical advantage of kernel methods in representation learning is not reflected: Reviewer FoeR

- Claim of scalability is not convincing based on the provided experiments: Reviewers FoeR, Tr1g, 9bGE

- No theoretical guarantees: Reviewer YmzL

**Reviewer Scores:**

The ratings are: 4,2,4, 4

In their rebuttal, the authors clarify that one key purpose of the proposed method is to reduce the number of landmarks in the  Nyström framework to provide scalability, so they did not work with much larger data sets.  They state that theoretical guarantees are outside the scope of the current work. Thus overall, the paper is missing one key advantage of kernel methods over NNs, namely the theoretical analysis that can be carried out.

Based on the rebuttal, I do not think the reviewers would have changed their scores substantially.

---

### Decision · Program_Chairs · 2026-01-26

Reject